# BooVI: Provably Efficient Bootstrapped Value Iteration

**Boyi Liu**[*]       **Qi Cai**[†]       **Zhuoran Yang**[‡]       **Zhaoran Wang**[§]

## Abstract

Despite the tremendous success of reinforcement learning (RL) with function approximation, efficient exploration remains a significant challenge, both practically and theoretically. In particular, existing theoretically grounded RL algorithms based on upper confidence bounds (UCBs), such as optimistic least-squares value iteration (LSVI), are often incompatible with practically powerful function approximators, such as neural networks. In this paper, we develop a variant of bootstrapped LSVI, namely BooVI, which bridges such a gap between practice and theory. Practically, BooVI drives exploration through (re)sampling, making it compatible with general function approximators. Theoretically, BooVI inherits the worst-case $\widetilde{O}(\sqrt{d^3 H^3 T})$-regret of optimistic LSVI in the episodic linear setting. Here $d$ is the feature dimension, $H$ is the episode horizon, and $T$ is the total number of steps.

## 1 Introduction

Reinforcement learning (RL) with function approximation demonstrates significant empirical success in a broad range of applications [e.g., 16, 38, 40, 45]. However, computationally and statistically efficient exploration of large and intricate state spaces remains a major barrier. Practically, the lack of temporally-extended exploration [31, 28, 30] in existing RL algorithms, e.g., deterministic policy gradient [39] and soft actor-critic [19], hinders them from solving more challenging tasks, e.g., Montezuma's Revenge in Atari Games [42, 17], in a more sample-efficient manner. Theoretically, it remains unclear how to design provably efficient RL algorithms with finite sample complexities or regrets that allow for practically powerful function approximators, e.g., neural networks.

There exist two principled approaches to efficient exploration in RL, namely optimism in the face of uncertainty [e.g., 4, 41, 21, 11, 12, 5, 23, 24] and posterior sampling [e.g., 29, 34, 33, 30, 27, 36].

- The optimism-based approach is often instantiated by incorporating upper confidence bounds (UCBs) into the estimated (action-)value functions as bonuses, which are used to direct exploration. In the tabular setting, the resulting algorithms, e.g., variants of upper confidence bound reinforcement learning (UCRL) [4, 21, 11, 5], are known to attain the optimal worst-case regret [32]. Beyond the tabular setting, it remains unclear how to construct closed-form UCBs in a principled manner for general function approximators, e.g., neural networks. The only exception is the linear setting [51, 52, 24, 8], where optimistic least-squares value iteration (LSVI) [24] is known to attain a near-optimal (with respect to $T = KH$) worst-case regret. However, the closed-form UCB therein is tailored to linear models [1, 9] rather than fully general-purpose.

---

[*]Northwestern University; boyiliu2018@u.northwestern.edu

[†]Northwestern University; qicai2022@u.northwestern.edu

[‡]Princeton University; zy6@princeton.edu

[§]Northwestern University; zhaoranwang@gmail.com

35th Conference on Neural Information Processing Systems (NeurIPS 2021).

- On the other hand, the posterior-based approach, which originates from Thompson sampling [43, 37], can be instantiated using randomized (action-)value functions [34, 30, 27, 36]. Unlike optimistic LSVI, the resulting algorithm, namely randomized LSVI, straightforwardly allows for general function approximators, as it only requires injecting random noise into the training data of LSVI. Ideally, such injected random noise does not depend on particular function approximators. In the tabular setting, randomized LSVI is known to attain near-optimal worst-case and Bayesian regrets [30, 36]. Meanwhile, in the linear setting, randomized LSVI is very recently shown to attain a near-optimal worst-case regret [54], which is only worse than that of optimistic LSVI by a factor of $\sqrt{dH}$. Here $d$ is the feature dimension and $H$ is the episode horizon. However, achieving such a regret requires a nontrivial modification of the injected random noise, which is tailored to linear models. Such a specialized modification diminishes the supposed practical advantage of randomized LSVI.

To bridge such a gap between practice and theory, we aim to answer the following question: Can we design an RL algorithm that simultaneously achieves the practical advantage of randomized LSVI and the theoretical guarantee of optimistic LSVI?

In this paper, we propose a variant of bootstrapped LSVI, namely BooVI, which combines the advantages of the optimism-based and posterior-based approaches. The key idea of BooVI is to use posterior sampling to implicitly construct an "optimistic version" of the estimated (action-)value functions in a data-driven manner. Unlike randomized LSVI, which samples from the posterior only once, e.g., by injecting random noise into the training data of LSVI, BooVI samples from the posterior multiple times, e.g., via the Langevin dynamics [48, 35]. Upon evaluating an action at a state, BooVI ranks the values of the randomized (action-)value functions sampled from the posterior in descending order and returns a top-ranked value, which can be shown to be approximately optimistic. Generally speaking, BooVI corresponds to bootstrapping the noise in the least-squares regression problem of LSVI. As a result, it can be viewed as a parametric bootstrap counterpart of the nonparametric bootstrap technique used in bootstrapped deep Q-networks (DQNs) [31, 30], which demonstrates significant empirical success in terms of exploration.

Compared with existing RL algorithms with function approximation, the advantage of BooVI is twofold:

- Practically, BooVI bypasses the explicit construction of the closed-form UCB in optimistic LSVI. Like randomized LSVI, BooVI straightforwardly allows for general function approximators, as it only requires injecting random noise into the training process of LSVI, e.g., via the Langevin dynamics.
- Theoretically, BooVI inherits the worst-case $\widetilde{O}(\sqrt{d^3 H^3 T})$-regret of optimistic LSVI in the linear setting. Here $d$ is the feature dimension, $H$ is the episode horizon, and $T$ is the total number of steps. Such a regret is better than the best known worst-case regret of randomized LSVI by a factor of $\sqrt{dH}$. More importantly, unlike the specialized variant of randomized LSVI studied in [54], BooVI can be applied to the linear setting "as is", without tailoring the injected random noise to linear models.

**More Related Work:** The idea of using posterior sampling to achieve optimism in a data-driven manner is previously studied in the linear bandit setting [26] and the tabular setting of RL [3]. In fact, our linear setting of RL covers the linear bandit setting as a special case, where the episode horizon $H$ is set to one, and the tabular setting of RL as another special case, where the feature mapping is the canonical basis of the state and action spaces. Although BooVI is practically applicable to general function approximators, our theoretical guarantee on its worst-case regret is only applicable to the linear setting. Despite the recent progress [49, 22, 13, 15], simultaneously achieving provable computational and statistical efficiency in exploration with general function approximators remains challenging. See, e.g., [2, 18] for the recent progress in the contextual bandit setting. Finally, we refer readers to [46, 53, 57, 55] for theories on the generalized models of linear MDPs, which all share similar (approximately) linear structure with the class of linear MDPs considered in the analysis of this paper. We expect our regret bound stays the same for those settings with linear structures, or admits an extra $O(\epsilon T)$ dependency over $T$ for those settings with approximately linear structure, where $\epsilon > 0$ describes the level of nonlinearity in the MDP. We also refer readers to [50] for sharper regret bounds in linear MDPs, which are out of the scope of this paper.

## 2 Background

In this section, we introduce the general problem setting.

**Episodic Markov Decision Process.** We consider the episodic Markov decision process represented by a tuple $(\mathcal{S}, \mathcal{A}, H, \mathcal{P}, r)$, where $\mathcal{S}$ is a compact state space, $\mathcal{A}$ is a finite action space with cardinality $A$, $H$ is the number of timesteps in each episode, $\mathcal{P} = \{\mathcal{P}_h\}_{h \in [H]}$ with $\mathcal{P}_h : \mathcal{S} \times \mathcal{S} \times \mathcal{A} \to [0, 1]$ for all $h \in [H]$ is the set of transition kernels, and $r = \{r_h\}_{h \in [H]}$ with $r_h : \mathcal{S} \times \mathcal{A} \to [0, 1]$ for all $h \in [H]$ is the set of reward functions.

At each timestep $h \in [H]$ of episode $k$, an agent at state $s_h^k \in \mathcal{S}$ with policy $\pi^k = \{\pi_h^k\}_{h \in [H]}$, where $\pi_h^k : \mathcal{A} \times \mathcal{S} \to \mathbb{R}$ for all $h \in [H]$, interacts with the environment by first taking an action $a_h^k$ with probability $\pi_h^k(a_h^k \mid s_h^k)$ and then receiving the corresponding reward $r_h^k = r_h(s_h^k, a_h^k)$.

We evaluate the performance of policy $\pi = \{\pi_h\}_{h \in [H]}$ starting from timestep $h$, and state-action pair $(s, a)$ using its action-value function $Q_h^\pi : \mathcal{S} \times \mathcal{A} \to \mathbb{R}$, which is defined as

$$Q_h^\pi(s, a) = \mathbb{E}_\pi \left[ \sum_{h'=h}^{H} r_{h'}(s_{h'}, a_{h'}) \, \middle| \, s_h = s, \ a_h = a \right],$$

for all $h \in [H]$. Correspondingly, the value function $V_h^\pi : \mathcal{S} \to \mathbb{R}$ of a policy $\pi$ is defined as

$$V_h^\pi(s) = \mathbb{E}_\pi \left[ \sum_{h'=h}^{H} r_{h'}(s_{h'}, a_{h'}) \, \middle| \, s_h = s \right] = \mathbb{E}_\pi \left[ Q_h^\pi(s_h, a) \, \middle| \, s_h = s \right],$$

for all $h \in [H]$. Here the expectation $\mathbb{E}_\pi[\cdot]$ is taken over the trajectory generated by $\pi$. Also, we let $Q_{H+1}^\pi \equiv 0$ and thus $V_{H+1}^\pi \equiv 0$. Furthermore, we denote by $V_h^*(s)$ the value function corresponding to the optimal policy $\pi^*$. Finally, for notational simplicity, we define $[\mathcal{P}_h V](s, a) = \mathbb{E}_{s' \sim \mathcal{P}_h(\cdot \mid s, a)}[V(s')]$, where $V : \mathcal{S} \to \mathbb{R}$ can be any function.

For any algorithm that generates a sequence of policies $\{\pi^k\}_{k \in [K]}$, we track its performance via the cumulative regret defined by

$$\text{Regret}(K) = \sum_{k=1}^{K} \left[ V_1^*(s_1^k) - V_1^{\pi^k}(s_1^k) \right],$$

where $K$ is the number of episodes. Here $s_1^k$ is the initial state of episode $k$, which is arbitrarily chosen at the start of the episode.

## 3 Bootstrapped Value Iteration

In this section, we introduce Bootstrapped Value Iteration (BooVI in Algorithm 1). For notational simplicity, we write $r_h(s_h^k, a_h^k)$ as $r_h^k$ throughout the rest of this paper. Also, in this paper, we write $\max\{\min\{\cdot, \cdot\}, 0\}$ as $\min\{\cdot, \cdot\}^+$, and denote by $\|\cdot\|$ the 2-norm for vectors.

**Least-Squares Value Iteration.** At timestep $h \in [H]$ of episode $k$, given the estimated action-value function $\widehat{Q}_{h+1}^k(s_{h+1}^\tau, a)$ for all $\tau \in [k-1]$ and $a \in \mathcal{A}$, let $\widehat{V}_{h+1}^k(s_{h+1}^\tau) = \max_{a \in \mathcal{A}} \widehat{Q}_{h+1}^k(s_{h+1}^\tau, a)$. Then, Least-Squares Value Iteration (LSVI) updates the parameter $\omega$ of the action-value function via

$$\widehat{\omega}_h^k \leftarrow \underset{\omega \in \mathbb{R}^d}{\operatorname{argmin}} \left\{ \lambda \cdot \|\omega\|^2 + \sum_{\tau=1}^{k-1} \left( r_h^\tau + \widehat{V}_{h+1}^k(s_{h+1}^\tau) - Q(s_h^\tau, a_h^\tau; \omega) \right)^2 \right\}, \tag{3.1}$$

where $Q(\cdot, \cdot; \cdot) : \mathcal{S} \times \mathcal{A} \times \mathbb{R}^d \to \mathbb{R}$ is the parameterization of action-value function with $\omega \in \mathbb{R}^d$ being its parameter, and $\lambda \geq 0$ is the regularization parameter. Although the deterministic parameter update by LSVI could exploit the historical data well, it has limited ability to address the exploration need in more challenging tasks.

**Bootstrapped Value Iteration.** To achieve guided exploration, we introduce BooVI (Algorithm 1), which uses bootstrapping to enforce the optimism of the estimated action-value function.

At the timestep $h$ of episode $k$, given the current data buffer $\mathcal{D}_h^k = \{(s_h^\tau, a_h^\tau, r_h^\tau, s_{h+1}^\tau)\}_{\tau \in [k-1]}$, which contains the data collected from the previous $k-1$ episodes, and the current bootstrapped state values $\{\widetilde{V}_{h+1}^k(s_{h+1}^\tau)\}_{\tau \in [k-1]}$, we define

$$y_h^\tau = r_h^\tau + \widetilde{V}_{h+1}^k(s_{h+1}^\tau), \quad \text{for all } \tau \in [k-1]. \tag{3.2}$$

With $p_0(\omega)$ as prior of $\omega$ and $p(y_h^\tau \mid (s_h^\tau, a_h^\tau), \omega)$ as the likelihood of $y_h^\tau$, the posterior of $\omega$ is given by

$$p\Big(\omega \,\Big|\, \{\widetilde{V}_{h+1}^k(s_{h+1}^\tau)\}_{\tau \in [k-1]}, \mathcal{D}_h^k\Big) \propto p_0(\omega) \cdot \prod_{\tau=1}^{k-1} p\big(y_h^\tau \mid (s_h^\tau, a_h^\tau), \omega\big). \tag{3.3}$$

In BooVI (Algorithm 1), we propose to sample repeatedly from such posterior for $N_k$ times, which is equivalent to sampling repeatedly from randomized action-value function since each sample $\omega_h^{k,i}$ corresponds to one randomized action-value function $Q(\cdot, \cdot; \omega_h^{k,i})$. Such a collection of posterior weights is later used to construct the bootstrapped action-value function in Algorithm 2.

To independently sample from the posterior in (3.3), we can consider, e.g., using the Langevin dynamics $\omega(t+1) \leftarrow \omega(t) + \Delta\omega(t)$, where

$$\Delta\omega(t) = \frac{\epsilon_t}{2} \cdot \Big( \nabla_\omega \log p_0\big(\omega(t)\big) + \sum_{\tau=1}^{k-1} \nabla_\omega \log p\big(y_h^\tau \mid (s_h^\tau, a_h^\tau), \omega(t)\big) \Big) + \eta_t, \tag{3.4}$$

where $\epsilon_t > 0$ is the stepsize, and $\eta_t \sim \mathcal{N}(0, \epsilon_t)$. We note here that, after running sufficient many iterations with suitable choices of stepsizes $\epsilon_t > 0$, the Langevin dynamics gives effectively independent parameter samples from the posterior in (3.3).

To instantiate the connection between the posterior (3.3) with the LSVI, we consider Gaussian prior and likelihood as an example. For the Gaussian prior $\omega \sim \mathcal{N}(0, I_d)$ and the Gaussian likelihood $p(y_h^\tau \mid (s_h^\tau, a_h^\tau), \omega) \propto \exp\{-(y_h^\tau - Q(s_h^\tau, a_h^\tau; \omega))^2/(2\sigma^2)\}$, the posterior of $\omega$ is given by

$$p\Big(\omega \,\Big|\, \{\widetilde{V}_{h+1}^k(s_{h+1}^\tau)\}_{\tau \in [k-1]}, \mathcal{D}_h^k\Big) \propto \exp\Big\{ -\frac{1}{2} \cdot \|\omega\|^2 - \frac{1}{2\sigma^2} \sum_{\tau=1}^{k-1} \big(y_h^\tau - Q(s_h^\tau, a_h^\tau; \omega)\big)^2 \Big\}, \tag{3.5}$$

where $\sigma > 0$ is an absolute constant. In this case, the maximum a posteriori (MAP) estimate of $\omega$ coincides with the weight estimate $\widehat{\omega}_h^k$ obtained by LSVI. Such observation suggests that sampling from the posterior distribution $p(\omega \mid \{\widetilde{V}_{h+1}^k(s_{h+1}^\tau)\}_{\tau \in [k-1]}, \mathcal{D}_h^k)$ would allows us to both achieve exploitation based on the available data, and generate noise for randomized exploration. More specifically, the $\Delta\omega(t)$ in (3.4) takes the form of

$$\Delta\omega(t) = \frac{\epsilon_t}{2} \cdot \Big[ -\omega(t) + \frac{1}{\sigma^2} \sum_{\tau=1}^{k-1} \big(y_h^\tau - Q\big(s_h^\tau, a_h^\tau; \omega(t)\big)\big) \cdot \nabla_\omega Q\big(s_h^\tau, a_h^\tau; \omega(t)\big) \Big] + \eta_t,$$

which can be viewed as using stochastic gradient descent to solve for the LSVI update (3.1) with $\lambda = \sigma^2$ and $\widehat{V}_{h+1}^k$ replaced by $\widetilde{V}_{h+1}^k$. See also in Lemma E.1 for the closed form of the posterior in a linear MDP setting with Gaussian prior and likelihood.

We would like to highlight that, although the theoretical guarantee in this paper is built upon the Gaussian prior and likelihood, the choices of prior and likelihood are flexible. For example, we can use uninformative prior for generalized linear model, and Gaussian process prior for kernel of overparameterized neural networks [6]. Moreover, as suggested by theoretical results on Langevin Dynamics [56] and illustrative experiments in Appendix F, the computational overhead caused by replacing the minimization of (3.1) with posterior sampling is mild.

---

**Algorithm 1** Bootstrapped Value Iteration (BooVI)

---

1: **Require:** MDP $(\mathcal{S}, \mathcal{A}, H, \mathcal{P}, r)$, action-value function parameterization $Q(\cdot, \cdot\,; \cdot) : \mathcal{S} \times \mathcal{A} \times \mathbb{R}^d \to \mathbb{R}$, number of episodes $K$, number of posterior weights $\{N_k\}_{k \in [K]}$, lower and upper bootstrapping ratios $\alpha, \beta \in (0, 1)$

2: Initialize the data buffer $\mathcal{D}_h^1 \leftarrow \{\}$ for $h \in [H]$

3: **For** episode $k = 1, \ldots, K$ **do**

4:     Set $N_{k,\alpha} \leftarrow \lceil \alpha \cdot N_k \rceil, N_{k,\beta} \leftarrow \lfloor \beta \cdot N_k \rfloor$, and $\omega_{H+1}^{k,i} \leftarrow 0$ for all $i \in [N_k]$

5:     Sample $n_k$ uniformly from $\{N_{k,\alpha}, N_{k,\alpha} + 1, \ldots, N_{k,\beta}\}$

6:     **For** timestep $h = H, \ldots, 1$ **do**

7:         Generate $\widetilde{Q}_{h+1}^k(s_{h+1}^\tau, a)$ using Algorithm 2 with weights $\{\omega_{h+1}^{k,i}\}_{i \in [N_k]}$ and parameter $n_k$ for all $a \in \mathcal{A}$ and $\tau \in [k-1]$

8:         $\widetilde{V}_{h+1}^k(s_{h+1}^\tau) \leftarrow \max_{a \in \mathcal{A}} \widetilde{Q}_{h+1}^k(s_{h+1}^\tau, a)$ for all $\tau \in [k-1]$

9:         Independently sample $\{\omega_h^{k,i}\}_{i \in [N_k]}$ from the posterior $p(\omega \,|\, \{\widetilde{V}_{h+1}^k(s_{h+1}^\tau)\}_{\tau \in [k-1]}, \mathcal{D}_h^k)$ defined in (3.3), e.g., using Langevin dynamics in (3.4)

10:     **end**

11:     **For** timestep $h = 1, \ldots, H$ **do**

12:         Generate $\widetilde{Q}_h^k(s_h^k, a)$ using Algorithm 2 with weights $\{\omega_h^{k,i}\}_{i \in [N_k]}$ and parameter $n_k$ for all $a \in \mathcal{A}$

13:         Take action $a_h^k \leftarrow \mathrm{argmax}_{a \in \mathcal{A}} \widetilde{Q}_h^k(s_h^k, a)$, and observe $r_h^k$ and $s_{h+1}^k$

14:         Update the data buffer $\mathcal{D}_h^{k+1} \leftarrow \mathcal{D}_h^k \cup \{(s_h^k, a_h^k, r_h^k, s_{h+1}^k)\}$

15:     **End**

16: **End**

---

The following Algorithm 2 serves as the critical building block of BooVI for enforcing the optimism of the estimated action-value. At episode $k$ in BooVI, for any $(s, a) \in \mathcal{S} \times \mathcal{A}$, Algorithm 2 ranks the estimated action-value functions corresponding to the posterior sample obtained in Line 1 of Algorithm 1 in ascending order. Then, in order to enforce the optimism of the estimated action-value function, Algorithm 2 resamples the $n_k$-th top-ranked value from the ordered estimated action-value functions in the manner of bootstrapping. Finally, to ensure sufficient optimism of the obtained bootstrapped action-value function, we extrapolate with a tunable parameter $\nu > 1$.

---

**Algorithm 2** Bootstrapping Action-Value Function

---

1: **Require:** Action-value function parameterization $Q(\cdot, \cdot\,; \cdot) : \mathcal{S} \times \mathcal{A} \times \mathbb{R}^d \to \mathbb{R}$, posterior sample $\{\omega_h^{k,i}\}_{i \in [N_k]}$, integer $n_k \in [N_k]$, extrapolation parameter $\nu > 1$, and state-action pair $(s, a)$

2: Compute $Q_h^{k,i}(s, a) \leftarrow Q(s, a; \omega_h^{k,i})$ for all $i \in [N_k]$

3: Set $\widehat{Q}_h^k(s, a) \leftarrow (1/N_k) \sum_{i=1}^{N_k} Q_h^{k,i}(s, a)$

4: Rank $\{Q_h^{k,i}(s, a)\}_{i \in [N_k]}$ in ascending order to obtain $\{Q_h^{k,(i)}(s, a)\}_{i \in [N_k]}$

5: **Output:** $\widetilde{Q}_h^k(s, a) \leftarrow \min\{(1 - \nu) \cdot \widehat{Q}_h^k(s, a) + \nu \cdot Q_h^{k,(n_k)}(s, a), H - h + 1\}^+$

---

**Remark 3.1** (Sample Efficiency and Computational Efficiency)**.** Here we clarify the differences between *sample efficiency* and *computational efficiency*. Throughout this paper, we refer *sample efficiency* to the learning efficiency with respect to the total number $T = KH$ of interactions with the environment. Such interactions can often time be very resource demanding. Thus, this paper seeks to provide an algorithm that aims to achieve the *sample efficiency* with respect to the number of such interactions. Although posterior weights are sampled in Algorithm 1, the efficiency of such posterior sampling process is categorized as *computational efficiency* since it is purely based on the current data buffer $\mathcal{D}_h^k$ and does not require any extra interaction with the environment.

**Remark 3.2** (Additional Computational Cost)**.** Here we discuss the computational cost of BooVI compared to LSVI. (i) Posterior sampling vs. least-square minimization: Consider the Langevin dynamics in (3.4). Since posterior sampling replaces and admits similar steps as least-square minimization, the computational cost should not differ significantly between LSVI and BooVI. (ii) Computing bootstrapped Q-function: To compute bootstrapped Q-function, at each state-action pair,

the extra sorting takes $O(N_k \log N_k)$ time complexity, which makes the action selection procedure $O(N_k \log N_k)$ times slower. In addition, keeping $N_k$ posterior weights takes $O(d \cdot N_k)$ memory. We note that, although the requirement on $N_k$ in our subsequent analysis is large, we should be able to use much smaller $N_k$ in practice due to the pessimistic nature of the analysis. See Appendix F for illustrative experiment results.

It is worth mentioning that BooVI is applicable to any specific parameterization $Q(\cdot, \cdot; \cdot) : \mathcal{S} \times \mathcal{A} \times \mathbb{R}^d \to \mathbb{R}$ of the action-value function and is fully data-driven. We also note here that the design of BooVI does not rely on the Gaussian prior/posterior or the Langevin dynamics sampling technique.

## 4 Main Results

While BooVI (Algorithm 1) is generally applicable to different parameterizations of the action-value function, we establish its regret for a class of linear MDPs.

**Linear Markov Decision Process.** We consider a class of MDPs, where the transition kernels and the reward functions are linear in the same feature mapping. Specifically, we have the following definition.

**Definition 4.1** (Linear MDP). An MDP $(\mathcal{S}, \mathcal{A}, H, \mathcal{P}, r)$ is called a linear MDP with feature mapping $\phi : \mathcal{S} \times \mathcal{A} \to \mathbb{R}^d$, if for any $h \in [H]$, there exists $d$ unknown signed measures $\mu_h = (\mu_h^{(1)}, \dots, \mu_h^{(d)})$ over $\mathcal{S}$ and an unknown vector $\theta_h \in \mathbb{R}^d$, such that for any $(s, a) \in \mathcal{S} \times \mathcal{A}$, we have

$$\mathcal{P}_h(\cdot \,|\, s, a) = \langle \phi(s, a), \mu_h(\cdot) \rangle, \quad r_h(s, a) = \langle \phi(s, a), \theta_h \rangle. \tag{4.1}$$

See [51, 52, 24] for examples of such a class of linear MDPs. Specifically, examples include tabular MDPs where $d = SA$ and the feature mapping is the canonical basis $\phi(s, a) = e_{(s,a)}$, and the simplex feature space where the feature space $\{\phi(s, a) \,|\, (s, a) \in \mathcal{S} \times \mathcal{A}\}$ is a subset of the $d$-dimensional simplex. See also [14, 44, 25] for related discussions on such a linear representation. For notational simplicity, we write the feature $\phi(s_h^k, a_h^k)$ as $\phi_h^k$ throughout the rest of this paper.

Based on Definition 4.1, without loss of generality, we make the following assumption.

**Assumption 4.2.** The MDP $(\mathcal{S}, \mathcal{A}, H, \mathcal{P}, r)$ is a linear MDP with $\|\phi(s, a)\| \leq 1$ for all $(s, a) \in \mathcal{S} \times \mathcal{A}$ and $\max\{\|\mu_h(\mathcal{S})\|, \|\theta_h\|\} \leq \sqrt{d}$ for all $h \in [H]$.

To motivate the linear parameterization of the action-value function in the subsequent section, we have the following proposition.

**Proposition 4.3** (Linear Action-Value Function, Proposition 2.3 in [24]). For a linear MDP, for any policy $\pi$ and $h \in [H]$, there exists $\omega_h^\pi \in \mathbb{R}^d$ such that for all $(s, a) \in \mathcal{S} \times \mathcal{A}$, we have $Q_h^\pi(s, a) = \langle \phi(s, a), \omega_h^\pi \rangle$.

**Regret of BooVI for Linear MDPs.** For a linear MDP, by Proposition 4.3, the parameterization $Q(\cdot, \cdot; \cdot) : \mathcal{S} \times \mathcal{A} \times \mathbb{R}^d \to \mathbb{R}$ in BooVI should take the form of $Q(s, a; \omega) = \langle \phi(s, a), \omega \rangle$, where $\omega \in \mathbb{R}^d$. Furthermore, when the prior and the likelihood are Gaussian, the posterior of $\omega$ is Gaussian distribution with mean being is the weight update by LSVI with chioce of $\lambda = \sigma^2$.

For any bootstrapping ratio $q \in (0, 1)$, we write $C_q = \Phi^{-1}(q)$ throughout the rest of this paper, where $\Phi(\cdot)$ is the cumulative distribution function of the standard Gaussian. Then, we have the following upper bound of regret of BooVI.

**Theorem 4.4.** Suppose that Assumption 4.2 holds, and that $d \geq 2$. We set $\sigma = 1$, $\nu = dH$, and fix a failure probability $p \in (0, 1]$. Then there exists a sequence of posterior sample sizes $\{N_k\}_{k \in [K]}$, and there exist absolute constants $c_\beta > c_\alpha > 0$, such that if $N_k = O(d^6 T^4 k / p^4)$ for all $k \in [K]$, $C_\alpha = c_\alpha \cdot \sqrt{\iota}$, and $C_\beta = c_\beta \cdot \sqrt{\iota}$, BooVI (Algorithm 1) with Gaussian posterior in (3.5) satisfies

$$\text{Regret}(K) = O\big(\sqrt{d^3 H^3 T \iota^2}\big)$$

with probability at least $1 - p$, where $T = KH$ and $\iota = \log(3dT/p)$.

*Proof.* See Section 5 for a proof sketch. $\qquad\square$

# 5 Proof Sketch

**Notation.** At episode $k$, given the number of posterior sample weights $N_k$ and given $n_k$ sampled in Line 1 of Algorithm 1, we define $C_k = \Phi^{-1}(n_k/N_k)$. At timestep $h$ of episode $k$, further given posterior weights $\{\omega_h^{k,i}\}_{i \in [N_k]}$, corresponding to the bootstrapped action-value function $\widetilde{Q}_h^k$ generated by Algorithm 2, we define the bootstrapped value function as

$$\widetilde{V}_h^k(s) = \max_{a \in \mathcal{A}} \widetilde{Q}_h^k(s, a). \tag{5.1}$$

Next, we define the mean bootstrapped action-value functions as

$$\overline{Q}_h^k(s, a) = \min\Big\{\mathbb{E}_\omega\big[(1 - \nu) \cdot \widehat{Q}_h^k(s, a) + \nu \cdot Q_h^{k,(n_k)}(s, a)\big], H - h + 1\Big\}^+, \tag{5.2}$$

where $\mathbb{E}_\omega[\,\cdot\,]$ is taken over $\omega$ with respect to the posterior $p(\omega \,|\, \{\widetilde{V}_{h+1}^k(s_{h+1}^\tau)\}_{\tau \in [k-1]}, \mathcal{D}_h^k)$ defined in (3.3). Correspondingly, we define the mean bootstrapped value function as

$$\overline{V}_h^k(s) = \max_{a \in \mathcal{A}} \overline{Q}_h^k(s, a). \tag{5.3}$$

Also, we denote by

$$\overline{\omega}_h^k = (\Lambda_h^k)^{-1} \sum_{\tau=1}^{k-1} y_h^\tau \cdot \phi_h^\tau \tag{5.4}$$

the mean of the posterior distribution $p(\omega \,|\, \{\widetilde{V}_{h+1}^k(s_{h+1}^\tau)\}_{\tau \in [k-1]}, \mathcal{D}_h^k)$, where $y_h^\tau$ is defined in (3.2). For a fixed failure probability $p \in (0, 1]$, we write $\iota = \log(3dT/p)$, where $T = KH$. Finally, we define matrix $\Lambda_h^k \in \mathbb{R}^{d \times d}$ by

$$\Lambda_h^k = \sum_{\tau=1}^{k-1} \phi_h^\tau (\phi_h^\tau)^\top + \sigma^2 \cdot I_d. \tag{5.5}$$

**Concentration Events.** Before proving Theorem 4.4, we first present two lemmas, each characterizing an event that is involved throughout the remaining proofs. The first lemma characterizes the concentration behavior of the bootstrapped action-value function $\widetilde{Q}_h^k$.

**Lemma 5.1.** Let $\sigma = 1$, $\nu = dH$, and $C_\beta = c_\beta \cdot \sqrt{\iota}$ for some $c_\beta > 0$. For a fixed failure probability $p \in (0, 1]$, we define $\mathcal{E}$ as the event that the condition

$$\big|\widetilde{Q}_h^k(s, a) - \overline{Q}_h^k(s, a)\big| \le \big(c_\beta/\sqrt{d} + 3\big) \cdot H\sqrt{d\iota/k}$$

is satisfied for all $(s, a, k, h) \in \mathcal{S} \times \mathcal{A} \times [K] \times [H]$. Then, there exists a sequence of posterior sample sizes $\{N_k\}_{k \in [K]}$ satisfying $N_k = O(d^6 T^4 k/p^4)$ for all $k \in [K]$, such that $P(\mathcal{E}) \ge 1 - p/3$.

*Proof.* See Appendix C for a detailed proof. □

The second lemma characterizes the concentration behavior of the mean bootstrapped state value function $\overline{V}_h^k$.

**Lemma 5.2.** Let $\sigma = 1$, $\nu = dH$, and $C_\beta = c_\beta \cdot \sqrt{\iota}$ for some constant $c_\beta > 0$. For a fixed failure $p \in (0, 1]$, we define $\mathcal{E}'$ as the event that the condition

$$\left\|\sum_{\tau=1}^{k-1} \phi_h^\tau \cdot \big(\overline{V}_{h+1}^k(s_{h+1}^\tau) - [\mathcal{P}_h \overline{V}_{h+1}^k](s_h^\tau, a_h^\tau)\big)\right\|_{(\Lambda_h^k)^{-1}} \le C \cdot dH\sqrt{\chi}$$

is satisfied for all $(k, h) \in [K] \times [H]$, where $\chi = \log[3(1+c_\beta)dT/p)]$. Then we have $\mathbb{P}(\mathcal{E}') \ge 1 - p/3$.

*Proof.* See Appendix D for a detailed proof. □

**Model Estimation Error.** With the events $\mathcal{E}$ and $\mathcal{E}'$ ready, we can proceed to characterize the model estimation error $\zeta_h^k : \mathcal{S} \times \mathcal{A} \to \mathbb{R}$ for all $(k, h) \in [K] \times [H]$ defined as

$$\zeta_h^k = \overline{Q}_h^k - \left(r_h + [\mathcal{P}_h \overline{V}_{h+1}^k]\right), \tag{5.6}$$

which can be comprehended as the estimation error of the model at timestep $h$ of episode $k$ induced by the mean bootstrapped value function $\overline{Q}_{h+1}^k$. We have the following lemma charaterizing the model estimation error $\zeta_h^k$ defined in (5.6).

**Lemma 5.3** (Model Estimation Error). *Let $d \geq 2$, $\nu = dH$, $\sigma = 1$, and $p \in (0, 1]$. Then there exists a sequence of posterior sample sizes $\{N_k\}_{k \in [K]}$, and there exist absolute constants $c_\beta > c_\alpha > 0$ such that, if $N_k = O(d^6 T^4 k / p^4)$ for all $k \in [K]$, $C_\alpha = c_\alpha \cdot \sqrt{\iota}$, and $C_\beta = c_\beta \cdot \sqrt{\iota}$, for the model estimation error $\zeta_h^k$ defined in (5.6), we under events $\mathcal{E}$ and $\mathcal{E}'$ that*

$$0 \leq \zeta_h^k(s, a) \leq (c_\alpha + c_\beta) \cdot dH\sqrt{\iota}\sqrt{\phi(s, a)^\top (\Lambda_h^k)^{-1} \phi(s, a)}$$

*for all $(s, a, k, h) \in \mathcal{S} \times \mathcal{A} \times [K] \times [H]$.*

*Proof.* Under Assumption 4.2, using (4.1) we have $[\mathcal{P}_h \overline{V}_{h+1}^k] = \int_{\mathcal{S}} \overline{V}_{h+1}^k(s') \cdot \langle \phi, \mu_h(\mathrm{d}s') \rangle$. Then, with slight abuse of notation, we denote by $\overline{V}_{h+1}^k(\mu_h) = \int_{\mathcal{S}} \overline{V}_{h+1}^k(s) \mu_h(\mathrm{d}s) \in \mathbb{R}^d$ and write

$$
\begin{aligned}
\mathcal{P}_h \overline{V}_{h+1}^k &= \left\langle \phi, (\Lambda_h^k)^{-1}(\Lambda_h^k) \overline{V}_{h+1}^k(\mu_h) \right\rangle \\
&= \left\langle \phi, (\Lambda_h^k)^{-1} \left( \sum_{\tau=1}^{k-1} \phi_h^\tau (\phi_h^\tau)^\top \overline{V}_{h+1}^k(\mu_h) \right) \right\rangle + \left\langle \phi, (\Lambda_h^k)^{-1} \overline{V}_{h+1}^k(\mu_h) \right\rangle \\
&= \left\langle \phi, (\Lambda_h^k)^{-1} \left( \sum_{\tau=1}^{k-1} \phi_h^\tau \cdot [\mathcal{P}_h \overline{V}_{h+1}^k](s_h^\tau, a_h^\tau) \right) \right\rangle + \left\langle \phi, (\Lambda_h^k)^{-1} \overline{V}_{h+1}^k(\mu_h) \right\rangle,
\end{aligned}
$$

where the second equality follows from (5.5). Thus, we have the following decomposition

$$\left| \langle \phi, \overline{\omega}_h^k \rangle - (r_h + \mathcal{P}_h \overline{V}_{h+1}^k) \right| \leq u_1 + u_2,$$

where $\overline{\omega}_h^k$ is defined in (5.4), and

$$u_1 = \left| \left\langle \phi, (\Lambda_h^k)^{-1} \left[ \sum_{\tau=1}^{k-1} \phi_h^\tau \cdot \left( \widetilde{V}_{h+1}^k(s_{h+1}^\tau) - [\mathcal{P}_h \overline{V}_{h+1}^k](s_h^\tau, a_h^\tau) \right) \right] \right\rangle \right|,$$

$$u_2 = \left| \left\langle \phi, (\Lambda_h^k)^{-1} \left( \sum_{\tau=1}^{k-1} r_h^\tau \cdot \phi_h^\tau - \overline{V}_{h+1}^k(\mu_h) \right) \right\rangle - r_h \right|.$$

In the sequel, we upper bound $u_1$ and $u_2$ separately.

**Upper bounding $u_1$:** First, we further decompose $u_1$ as

$$\left\langle \phi, (\Lambda_h^k)^{-1} \left\{ \sum_{\tau=1}^{k-1} \phi_h^\tau \cdot \left[ \left( \widetilde{V}_{h+1}^k(s_{h+1}^\tau) - \overline{V}_{h+1,}^k(s_{h+1}^\tau) \right) + \left( \overline{V}_{h+1}^k(s_{h+1}^\tau) - [\mathcal{P}_h \overline{V}_{h+1}^k](s_h^\tau, a_h^\tau) \right) \right] \right\} \right\rangle,$$

applying the Cauchy-Schwartz inequality to which we obtain

$$u_1 \leq \left\langle \phi, (\Lambda_h^k)^{-1} \left[ \sum_{\tau=1}^{k-1} \phi_h^\tau \cdot \left( \widetilde{V}_{h+1}^k(s_{h+1}^\tau) - \overline{V}_{h+1}^k(s_{h+1}^\tau) \right) \right] \right\rangle \tag{5.7}$$

$$+ \sqrt{\phi^\top (\Lambda_h^k)^{-1} \phi} \cdot \left\| \sum_{\tau=1}^{k-1} \phi_h^\tau \cdot \left( \overline{V}_{h+1}^k(s_h^\tau, a_h^\tau) - [\mathcal{P}_h \overline{V}_{h+1}^k](s_{h+1}^\tau) \right) \right\|_{(\Lambda_h^k)^{-1}}.$$

Now we proceed to upper bound the two terms on the right-hand side of (5.7) in the following. For the first term, since by Lemma 5.1 we have under event $\mathcal{E}$ that $|\widetilde{V}_{h+1}^k(s_{h+1}^\tau) - \overline{V}_{h+1}^k(s_{h+1}^\tau)| \leq (c_\beta/\sqrt{d} + 3) \cdot H\sqrt{d\iota/k}$, we have

$$\left| \left\langle \phi, (\Lambda_h^k)^{-1} \left[ \sum_{\tau=1}^{k-1} \phi_h^\tau \cdot \left( \widetilde{V}_{h+1}^k(s_{h+1}^\tau) - \overline{V}_{h+1}^k(s_{h+1}^\tau) \right) \right] \right\rangle \right| \leq (c_\beta/\sqrt{d} + 3) \cdot H\sqrt{d\iota/k} \sum_{\tau=1}^{k-1} \phi^\top (\Lambda_h^k)^{-1} \phi_h^\tau.$$

$$(5.8)$$

By the Cauchy-Schwartz inequality and Lemma D.1 in [24] (see Appendix E), we have

$$\sum_{\tau=1}^{k-1} \phi^\top (\Lambda_h^k)^{-1} \phi_h^\tau \leq \left[ \left( \sum_{\tau=1}^{k-1} \phi^\top (\Lambda_h^k)^{-1} \phi_h^\tau \right) \cdot \left( \sum_{\tau=1}^{k-1} \phi^\top (\Lambda_h^k)^{-1} \phi \right) \right]^{1/2} \leq \sqrt{d\iota/k} \cdot \sqrt{dk} \sqrt{\phi^\top (\Lambda_h^k)^{-1} \phi},$$

taking which into (5.8) and gives

$$\left| \left\langle \phi, (\Lambda_h^k)^{-1} \left[ \sum_{\tau=1}^{k-1} \phi_h^\tau \cdot \left( \widetilde{V}_{h+1}^k(s_{h+1}^\tau) - \overline{V}_{h+1}^k(s_{h+1}^\tau) \right) \right] \right\rangle \right| \leq (c_\beta/\sqrt{d} + 3) \cdot dH \sqrt{\iota} \sqrt{\phi^\top (\Lambda_h^k)^{-1} \phi}. \tag{5.9}$$

Taking (5.9) into (5.7) and applying Lemma 5.2 to the second term on the right-hand side of (5.7), we have under the events $\mathcal{E}$ and $\mathcal{E}'$ that

$$u_1 \leq \left[ (c_\beta/\sqrt{d} + 3) \cdot dH \sqrt{\iota} + C \cdot dH \sqrt{\chi} \right] \cdot \sqrt{\phi^\top (\Lambda_h^k)^{-1} \phi}. \tag{5.10}$$

**Upper bounding $u_2$:** By (4.1), we have

$$u_2 = \left| \left\langle \phi, (\Lambda_h^k)^{-1} \left( \theta_h + \overline{V}_{h+1}^k(\mu_h) \right) \right\rangle \right|.$$

Then, by the Cauchy-Schwartz inequality, we further obtain

$$u_2 \leq \sqrt{\phi^\top (\Lambda_h^k)^{-1} \phi} \cdot \left( \left\| \overline{V}_{h+1}^k(\mu_h) \right\|_{(\Lambda_h^k)^{-1}} + \left\| \theta_h \right\|_{(\Lambda_h^k)^{-1}} \right)$$

$$\leq \sqrt{\phi^\top (\Lambda_h^k)^{-1} \phi} \cdot \left( \left\| \overline{V}_{h+1}^k(\mu_h) \right\| + \left\| \theta_h \right\| \right) \leq 2H\sqrt{d} \cdot \sqrt{\phi^\top (\Lambda_h^k)^{-1} \phi}, \tag{5.11}$$

where the second inequality follows from $\Lambda_h^k \succeq I_d$, and the last inequality follows from $\|\theta_h\| \leq \sqrt{d}$ in Assumption (4.2) as well as the fact that $\overline{V}_{h+1}^k(s) \leq H - h \leq H - 1$ for all $s \in \mathcal{S}$.

Combining (5.10) and (5.11), we obtain under event $\mathcal{E}$ that

$$\left| \langle \phi, \overline{\omega}_h^k \rangle - (r_h + \mathcal{P}_h \overline{V}_{h+1}^k) \right| \Big/ \sqrt{\phi^\top (\Lambda_h^k)^{-1} \phi} \tag{5.12}$$

$$\leq \left[ (c_\beta/\sqrt{d} + 3) \cdot dH \sqrt{\iota} + C \cdot dH \sqrt{\chi} + 2H\sqrt{d} \right]$$

$$\leq \left[ (c_\beta/\sqrt{d} + 5) \cdot dH \sqrt{\iota} + C \cdot dH \sqrt{\chi} \right] = C' \cdot dH \sqrt{\iota},$$

where $C' > 0$ is an absolute constant. Next, we need to find an absolute constant $c_\beta > 0$ such that $C' \cdot \sqrt{\iota} = (c_\beta/\sqrt{d} + 5) \cdot \sqrt{\iota} + C \cdot \sqrt{\iota + \log(1 + c_\beta)} < c_\beta \cdot \sqrt{\iota}$. Note that $d \geq 2$ and $\iota \geq \log 2$, it suffices to pick a $c_\beta > 0$ such that

$$C \cdot \sqrt{\log 2 + \log(1 + c_\beta)} < \left[ (1 - 1/\sqrt{2}) c_\beta - 5 \right] \cdot \sqrt{\log 2}, \tag{5.13}$$

which must exist as the left hand side grows logarithmically in $c_\beta$ and the right-hand side grows linearly in $c_\beta$. For $c_\beta > 0$ satisfying (5.13), we pick any $c_\alpha > 0$ such that $C' \leq c_\alpha < c_\beta$ and let $C_\alpha = c_\alpha \cdot \sqrt{\iota}$. By (5.12) and $r_h + [\mathcal{P}_h \overline{V}_{h+1}^k] \leq H - h + 1$, we obtain under events $\mathcal{E}$ and $\mathcal{E}'$ that

$$\zeta_h^k = \min \left\{ \langle \phi, \overline{\omega}_h^k \rangle + C_k \cdot \nu \cdot \sqrt{\phi^\top (\Lambda_h^k)^{-1} \phi}, H - h + 1 \right\}^+ - (r_h + \mathcal{P}_h \overline{V}_{h+1}^k)$$

$$\geq \min \left\{ (c_\alpha - C') \cdot dH \sqrt{\iota} \sqrt{\phi^\top (\Lambda_h^k)^{-1} \phi}, 0 \right\} \geq 0.$$

On the other hand, by (5.12), we also have under events $\mathcal{E}$ and $\mathcal{E}'$ that

$$\zeta_h^k \leq \langle \phi, \overline{\omega}_h^k \rangle - (r_h + \mathcal{P}_h \overline{V}_{h+1}^k) + C_\beta \cdot \nu \cdot \sqrt{\phi^\top (\Lambda_h^k)^{-1} \phi} \leq (c_\alpha + c_\beta) \cdot dH \sqrt{\iota} \sqrt{\phi^\top (\Lambda_h^k)^{-1} \phi}.$$

Therefore, we finish the proof of Lemma 5.3. $\qquad \square$

As results of Lemma 5.3, we have the following two lemmas characterizing the optimism and the cumulative estimation error of the mean bootstrapped value function $\overline{V}_h^k$, respectively.

**Lemma 5.4** (Optimistic Random Value Function). Let $\sigma = 1$ and $\nu = dH$. There there exists a sequence of posterior sample sizes $\{N_k\}_{k \in [K]}$, and there exist absolute constants $c_\beta > c_\alpha > 0$, such that if $N_k = O(d^6 T^4 k/p^4)$ for all $k \in [K]$, $C_\alpha = c_\alpha \cdot \sqrt{\iota}$, and $C_\beta = c_\beta \cdot \sqrt{\iota}$, we have under the events $\mathcal{E}$ and $\mathcal{E}'$ that

$$\sum_{k=1}^{K} \big[V_1^*(s_1^k) - \overline{V}_1^k(s_1^k)\big] \leq 0.$$

*Proof.* See Appendix D for a detailed proof. □

**Lemma 5.5** (Cumulative Estimation Error). Let $\sigma = 1$, $\nu = dH$, and $p \in (0, 1]$. There exists a sequence of posterior sample sizes $\{N_k\}_{k \in [K]}$, and there exist absolute constants $c_\beta > c_\alpha > 0$, such that if $N_k = O(d^6 T^4 k/p^4)$ for all $k \in [K]$, $C_\alpha = c_\alpha \cdot \sqrt{\iota}$, and $C_\beta = c_\beta \cdot \sqrt{\iota}$, we have under the events $\mathcal{E}$ and $\mathcal{E}'$ that

$$\sum_{k=1}^{K} \big[\overline{V}_1^k(s_1^k) - V_1^{\pi^k}(s_1^k)\big] \leq \sqrt{18TH^2 \cdot \log(3/p)} + 4(c_\beta/\sqrt{d} + 3) \cdot H^2 \sqrt{dK\iota}$$
$$+ (c_\alpha + c_\beta) \cdot dH^2 \sqrt{2dK\iota^2},$$

with probability at least $1 - p/3$.

*Proof.* See Appendix D for a detailed proof. □

Finally, the regret bound of BooVI for the class of linear MDPs can be established as a consequence of the optimism (Lemmas 5.4) and the upper bound of the cumulative estimation error (Lemma 5.5) of the bootstrapped value function.

*Proof of Theorem 4.4.* First, recall that we have $\overline{V}_h^k(s)$ defined in (5.3). We have the regret decomposition

$$\text{Regret}(K) = \sum_{k=1}^{K} \big[V_1^*(s_1^k) - \overline{V}_1^k(s_1^k)\big] + \sum_{k=1}^{K} \big[\overline{V}_1^k(s_1^k) - V_1^{\pi^k}(s_1^k)\big]. \tag{5.14}$$

Applying Lemmas 5.4 and 5.5 to (5.14), we obtain under the events $\mathcal{E}$ and $\mathcal{E}'$ that

$$\text{Regret}(K) \leq 0 + \sqrt{18TH^2 \cdot \log(3/p)} + 4(c_\beta/\sqrt{d} + 3) \cdot H^2 \sqrt{dK\iota}$$
$$+ (c_\alpha + c_\beta) \cdot dH^2 \sqrt{2dK\iota^2}$$
$$= O\big(\sqrt{d^3 H^3 T\iota^2}\big) \tag{5.15}$$

with probability at least $1 - p/3$. Finally, by Lemmas 5.1 and 5.2, we have with at least probability $1 - p/3 - p/3 = 1 - 2p/3$ that the events $\mathcal{E}$ and $\mathcal{E}'$ hold simultaneously. Thus, we have (5.15) hold with probability at least $1 - p$, which concludes the proof. □

## 6 Conclusion

The cost of collecting data via online experiments is often prohibitive compared to collecting offline data, e.g., via posterior sampling. Thus, designing online learning algorithms that provably explores the environment in a data-driven manner is essential. Moreover, applicability to general environments is critical since reinforcement learning tasks are getting more chanlleging and complex recently. We aspire to motivate the design of novel reinforcement learning algorithms that utilize cheaper data to boost online sample efficiency. Our algorithm and analysis serve as a step towards developing general applicable and provable sample-efficient reinforcement learning. While this paper is mainly on algorithmic and theoretical aspects of the bootstrapping idea in RL, it would be interesting to see the empirical strength of BooVI in more challenging RL environments. We leave this part to our future work.

## Acknowledgement

Zhaoran Wang acknowledges National Science Foundation (Awards 2048075, 2008827, 2015568, 1934931), Simons Institute (Theory of Reinforcement Learning), Amazon, J.P. Morgan, and Two Sigma for their supports.

Zhuoran Yang acknowledges Simons Institute (Theory of Reinforcement Learning).

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
