## A  Proof of Lemma 5.4

With Lemma 5.3, the proof of Lemma 5.4 mainly follows that of Lemma B.5 in [24]. For self-containedness, we lay out the proof of Lemma 5.4 in the following.

*Proof.* By the definition of $\zeta_h^k$ in (5.6), we have

$$
\begin{aligned}
Q_h^*(s,a) - \overline{Q}_h^k(s,a) &= Q_h^*(s,a) - \big(r_h + [\mathcal{P}_h \overline{V}_{h+1}^k]\big)(s,a) - \zeta_h^k(s,a) \\
&= \big(r_h + [\mathcal{P}_h V_{h+1}^*]\big)(s,a) - \big(r_h + [\mathcal{P}_h \overline{V}_{h+1}^k]\big)(s,a) - \zeta_h^k(s,a) \\
&= \big[\mathcal{P}_h(V_{h+1}^* - \overline{V}_{h+1}^k)\big](s,a) - \zeta_h^k(s,a),
\end{aligned}
$$

applying Lemma 5.3 to which we obtain under the events $\mathcal{E}$ and $\mathcal{E}'$ that, for all $(s,a,k,h) \in \mathcal{S} \times \mathcal{A} \times [K] \times [H]$,

$$
Q_h^*(s,a) - \overline{Q}_h^k(s,a) \le \big[\mathcal{P}_h(V_{h+1}^* - \overline{V}_{h+1}^k)\big](s,a).
$$

Note that by definition we have $Q_{H+1}^* \equiv \overline{Q}_{H+1}^k \equiv 0$, which gives $V_{H+1}^* \equiv \overline{V}_{H+1}^k \equiv 0$. Thus, we have for all $(s,a) \in \mathcal{S} \times \mathcal{A}$ that

$$
Q_H^*(s,a) - \overline{Q}_H^k(s,a) \le 0,
$$

which gives $V_H^*(s) - \overline{V}_H^k(s) \le 0$ for all $s \in \mathcal{S}$. Thus, we have

$$
Q_{H-1}^*(s,a) - \overline{Q}_{H-1}^k(s,a) \le \big[\mathcal{P}_h(V_H^* - \overline{V}_H^k)\big](s,a) \le 0.
$$

Finally, by induction, we obtain $V_1^*(s_1) - \overline{V}_1^k(s_1) \le 0$ under events $\mathcal{E}$ and $\mathcal{E}'$, summing up which for all $k \in [K]$ concludes the proof. $\qquad\square$

## B  Proof of Lemma 5.5

First, we present the following lemma that decomposes the term $\overline{V}_{1,\beta}^k(s_1^k) - V_1^{\pi^k}(s_1^k)$.

**Lemma B.1.** For $(s,a,k,h) \in \mathcal{S} \times \mathcal{A} \times [K] \times [H]$, let $\psi_h^k(s_h^k,a_h^k) = \overline{V}_h^k(s_h^k) - \overline{Q}_h^k(s_h^k,a_h^k)$ and

$$
\begin{aligned}
\xi_h^k(a_h^k,s_h^k,s_{h+1}^k) &= Q_h^{\pi^k}(s_h^k,a_h^k) - V_h^{\pi^k}(s_h^k) \\
&\quad + \big[\mathcal{P}_h(\overline{V}_{h+1}^k - V_{h+1}^{\pi^k})\big](s_h^k,a_h^k) - (\overline{V}_{h+1}^k - V_{h+1}^{\pi^k})(s_{h+1}^k). \tag{B.1}
\end{aligned}
$$

We have

$$
\overline{V}_1^k(s_1^k) - V_1^{\pi^k}(s_1^k) = \sum_{h=1}^H \xi_h^k(s_h^k,a_h^k,s_{h+1}^k) + \sum_{h=1}^H \psi_h^k(s_h^k,a_h^k) + \sum_{h=1}^H \zeta_h^k(s_h^k,a_h^k), \tag{B.2}
$$

where $\zeta_h^k$ is defined in (5.6)

*Proof.* In this proof, we first consider timesteps within an episode $k \in [K]$. First, by definition of $\zeta_h^k$ in (5.6), we have

$$
\begin{aligned}
\zeta_h^k(s_h^k,a_h^k) &= (\overline{Q}_h^k - Q_h^{\pi^k})(s_h^k,a_h^k) + \Big[Q_h^{\pi^k}(s_h^k,a_h^k) - \big(r_h + [\mathcal{P}_h \overline{V}_{h+1}^k]\big)(s_h^k,a_h^k)\Big] \\
&= (\overline{Q}_h^k - Q_h^{\pi^k})(s_h^k,a_h^k) + \Big[\big(r_h + [\mathcal{P}_h V_{h+1}^{\pi^k}]\big)(s_h^k,a_h^k) - \big(r_h + [\mathcal{P}_h \overline{V}_{h+1}^k]\big)(s_h^k,a_h^k)\Big] \\
&= (\overline{Q}_h^k - Q_h^{\pi^k})(s_h^k,a_h^k) - \big[\mathcal{P}_h(\overline{V}_{h+1}^k - V_{h+1}^{\pi^k})\big](s_h^k,a_h^k),
\end{aligned}
$$

where the second line follows from Bellman equation. Then we have for all $(h, k) \in [H] \times [K]$ that

$$\overline{V}_h^k(s_h^k) - V_h^{\pi^k}(s_h^k)$$
$$= (\overline{V}_h^k(s_h^k) - V_h^{\pi^k}(s_h^k)) - (\overline{Q}_h^k(s_h^k, a_h^k) - Q_h^{\pi^k}(s_h^k, a_h^k)) + [\mathcal{P}_h(\overline{V}_{h+1}^k - V_{h+1}^{\pi^k})](s_h^k, a_h^k) + \zeta_h^k(s_h^k, a_h^k)$$
$$= \underbrace{Q_h^{\pi^k}(s_h^k, a_h^k) - V_h^{\pi^k}(s_h^k) + [\mathcal{P}_h(\overline{V}_{h+1}^k - V_{h+1}^{\pi^k})](s_h^k, a_h^k) - (\overline{V}_{h+1}^k - V_{h+1}^{\pi^k})(s_{h+1}^k)}_{= \xi_h^k(s_h^k, a_h^k, s_{h+1}^k)}$$
$$+ \underbrace{\overline{V}_h^k(s_h^k) - \overline{Q}_h^k(s_h^k, a_h^k)}_{= \psi_h^k(s_h^k, a_h^k)} + \overline{V}_{h+1}^k(s_{h+1}^k) - V_{h+1}^{\pi^k}(s_{h+1}^k) + \zeta_h^k(s_h^k, a_h^k). \tag{B.3}$$

Applying (B.3) recursively for all $h \in [H]$ and recalling that $\overline{Q}_{H+1}^k(s, a) = Q_{H+1}^{\pi^k}(s, a) = 0$ for all $(s, a) \in \mathcal{S} \times \mathcal{A}$, we finish the proof. $\qquad\square$

Next, we bound the summation of $\xi_h^k(s_h^k, a_h^k, s_{h+1}^k)$ on the right-hand side of (B.2) in the following lemma.

**Lemma B.2.** Suppose that event $\mathcal{E}$ holds. For $\xi_h^k(s_h^k, a_h^k, s_{h+1}^k)$ defined in (B.1), we have with probability at least $1 - p/3$ that

$$\sum_{k=1}^{K} \sum_{h=1}^{H} \xi_h^k(s_h^k, a_h^k, s_{h+1}^k) \leq \sqrt{18TH^2 \cdot \log(3/p)}.$$

*Proof.* First, we define a $\sigma$-algebra that starts with $\mathcal{F}_1^1 = \{s_1^1\}$ and is then recursively constructed by

$$\mathcal{F}_h^k = \mathcal{F}_{h-1}^k \cup \{(a_{h-1}^k, r_{h-1}^k, s_h^k)\}, \quad 2 \leq h \leq H \tag{B.4}$$

and

$$\mathcal{F}_1^{k+1} = \mathcal{F}_H^k \cup \{\omega_h^{k+1,i}\}_{h \in [H], i \in [N_k]} \cup \{s_1^{k+1}\} \cup \{n_{k+1}\}. \tag{B.5}$$

Note that given the posterior sample $\{\omega_h^{k,i}\}_{h \in [H], i \in [N_k]}$ and $n_k$, the action-value function $Q_h^{\pi^k}$ and the bootstrapped action-value function $\widetilde{Q}_h^k$, as functions mapping from $\mathcal{S} \times \mathcal{A}$ to $\mathbb{R}$, are known. Thus, we have

$$\mathbb{E}\left[\xi_h^k(s_h^k, a_h^k, s_{h+1}^k) \,\middle|\, \mathcal{F}_h^k\right] = 0,$$

which means that the sequence $\{\xi_h^k(s_h^k, a_h^k)\}_{k \in [K], h \in [H]}$ is a martingale sequence adapted to the filtration $\{\mathcal{F}_h^k\}_{k \in [K], h \in [H]}$. Furthermore, by the definition of $\overline{Q}_h^k$ and $\overline{V}_h^k$ in (5.2) and (5.3), we know that the martingale sequence is bounded by $3H$. Therefore, by the Azuma-Hoeffding inequality, we obtain for any $t > 0$ that

$$\mathbb{P}\left(\sum_{k=1}^{K} \sum_{h=1}^{H} \xi_h^k(s_h^k, a_h^k, s_{h+1}^k) > t\right) \leq \exp\left(\frac{-t^2}{2T \cdot 9H^2}\right), \tag{B.6}$$

where $T = KH$. Taking $t = \sqrt{18TH^2 \log(3/p)}$ in (B.6), we finish the proof. $\qquad\square$

We have the following lemma bounding the summation of $\psi_h^k(s_h^k, a_h^k)$ defined in Lemma B.1.

**Lemma B.3.** Suppose that event $\mathcal{E}$ holds. For $\psi_h^k(s_h^k, a_h^k)$ defined in Lemma (B.1), we have

$$\sum_{k=1}^{K} \sum_{h=1}^{H} \psi_h^k(s_h^k, a_h^k) \leq 4(c_\beta/\sqrt{d} + 3) \cdot H^2 \sqrt{dK\iota}.$$

*Proof.* First, we make the decomposition

$$\overline{V}_h^k(s_h^k) - \overline{Q}_h^k(s_h^k, a_h^k) = \overline{V}_h^k(s_h^k) - \max_{a \in \mathcal{A}} \widetilde{Q}_h^k(s_h^k, a) + \max_{a \in \mathcal{A}} \widetilde{Q}_h^k(s_h^k, a) - \overline{Q}_h^k(s_h^k, a_h^k). \tag{B.7}$$

By the definition of $\overline{V}_h^k$ in (5.3), we have

$$\overline{V}_h^k(s_h^k) - \max_{a \in \mathcal{A}} \widetilde{Q}_h^k(s_h^k, a) = \max_{a \in \mathcal{A}} \overline{Q}_h^k(s_h^k, a) - \max_{a \in \mathcal{A}} \widetilde{Q}_h^k(s_h^k, a)$$

$$\leq \max_{a \in \mathcal{A}} \left| \overline{Q}_h^k(s_h^k, a) - \widetilde{Q}_h^k(s_h^k, a) \right|. \tag{B.8}$$

Meanwhile, since $a_h^k$ is the greedy action with respect to $\widetilde{Q}_h^k(s_h^k, a)$, we have

$$\max_{a \in \mathcal{A}} \widetilde{Q}_h^k(s_h^k, a) - \overline{Q}_h^k(s_h^k, a_h^k) = \widetilde{Q}_h^k(s_h^k, a_h^k) - \overline{Q}_h^k(s_h^k, a_h^k). \tag{B.9}$$

Taking (B.8) and (B.9) into (B.7), we obtain

$$\overline{V}_h^k(s_h^k) - \overline{Q}_h^k(s_h^k, a_h^k) \leq \max_{a \in \mathcal{A}} \left| \overline{Q}_h^k(s_h^k, a) - \widetilde{Q}_h^k(s_h^k, a) \right| + \widetilde{Q}_h^k(s_h^k, a_h^k) - \overline{Q}_h^k(s_h^k, a_h^k) \tag{B.10}$$

$$\leq 2(c_\beta/\sqrt{d} + 3) \cdot H\sqrt{d\iota/k}, \tag{B.11}$$

where the second inequality holds by the definition of the event $\mathcal{E}$ in Lemma 5.1. Finally, summing up (B.10) for all $h \in [H]$ and $k \in [K]$, we have

$$\sum_{k=1}^K \sum_{h=1}^H \overline{V}_h^k(s_h^k) - \overline{Q}_h^k(s_h^k, a_h^k) \leq 2(c_\beta/\sqrt{d} + 3) \cdot H^2 \sqrt{d\iota} \sum_{k=1}^K 1/\sqrt{k}$$

$$\leq 2(c_\beta/\sqrt{d} + 3) \cdot H^2 \sqrt{d\iota} \cdot 2\sqrt{K} = 4(c_\beta/\sqrt{d} + 3) \cdot H^2 \sqrt{dK\iota}.$$

Therefore, we conclude the proof of Lemma B.3. $\qquad\qquad\square$

Now we are ready to prove Lemma 5.5.

*Proof of Lemma 5.5.* First, summing (B.2) for all $k \in [K]$ we have

$$\sum_{k=1}^K \left[ \overline{V}_1^k(s_1^k) - V_1^{\pi^k}(s_1^k) \right] = \sum_{k=1}^K \sum_{h=1}^H \xi_h^k(s_h^k, a_h^k, s_{h+1}^k) + \sum_{k=1}^K \sum_{h=1}^H \psi_h^k(s_h^k, a_h^k) + \sum_{k=1}^K \sum_{h=1}^H \zeta_h^k(s_h^k, a_h^k).$$

By Lemmas B.2 and 5.2, under the event $\mathcal{E}$, we further have with probability at least $1 - p/3$ that

$$\sum_{k=1}^K \left[ \overline{V}_1^k(s_1^k) - V_1^{\pi^k}(s_1^k) \right] \leq \sqrt{18TH^2 \cdot \log(3/p)} + 4(c_\beta/\sqrt{d} + 3) \cdot H^2 \sqrt{dK\iota}$$

$$+ (c_\alpha + c_\beta) \cdot dH\sqrt{\iota} \sum_{h=1}^H \sum_{k=1}^K \sqrt{(\phi_h^k)^\top (\Lambda_h^k)^{-1} \phi_h^k}. \tag{B.12}$$

Next, we bound the third term in (B.12). By the Cauchy-Schwartz inequality, we have

$$\sum_{k=1}^K \sqrt{(\phi_h^k)^\top (\Lambda_h^k)^{-1} \phi_h^k} \leq \sqrt{K} \cdot \left[ \sum_{k=1}^K (\phi_h^k)^\top (\Lambda_h^k)^{-1} \phi_h^k \right]^{1/2} \tag{B.13}$$

$$\leq \sqrt{K} \cdot \left[ 2\log\left( \frac{\det(\Lambda_h^K)}{\det(\Lambda_h^0)} \right) \right]^{1/2} \leq \sqrt{K} \cdot \left[ 2d \cdot \log(1 + K) \right]^{1/2},$$

where the second inequality follows from Lemma E.4, and the last inequality follows from $I_d \preceq \Lambda_h^k \preceq (1 + k) \cdot I_d$. Further summing (B.13) up for all $h \in [H]$ and using $\log(1 + K) \leq \iota$, we have

$$\sum_{k=1}^K \sum_{h=1}^H \sqrt{(\phi_h^k)^\top (\Lambda_h^k)^{-1} \phi_h^k} \leq H\sqrt{2dK} \cdot \left[ 2d \cdot \log(1 + K) \right]^{1/2} \leq H\sqrt{2dK\iota},$$

taking which back into (B.12) gives

$$\sum_{k=1}^K \left[ \overline{V}_1^k(s_1^k) - V_1^{\pi^k}(s_1^k) \right] \leq \sqrt{18TH^2 \cdot \log(3/p)} + 4(c_\beta/\sqrt{d} + 3) \cdot H^2 \sqrt{dK\iota}$$

$$+ (c_\alpha + c_\beta) \cdot dH^2 \sqrt{2dK\iota^2}.$$

Thus, we conclude the proof of Lemma 5.5. $\qquad\qquad\square$

# C  Proof of Lemma 5.1

Prior to proving the uniform concentration guarantee over all $(s, a) \in \mathcal{S} \times \mathcal{A}$ in Lemma 5.1, we first present the following concentration lemma for a fixed state-action pair $(s, a)$. We drop the truncations at $0$ and $H - h + 1$ for the bootstrapped action-value function in all the proofs in this section, since for any random variable $z \in \mathbb{R}$ with mean $\overline{z}$,

$$\mathbb{P}\Big(\big|\min\{z, H - h + 1\}^+ - \min\{\overline{z}, H - h + 1\}^+\big| > \nu\epsilon\Big) \leq \mathbb{P}\big(|z - \overline{z}| > \nu\epsilon\big).$$

**Lemma C.1.** For fixed $(s, a, k, h) \in \mathcal{S} \times \mathcal{A} \times [K] \times [H]$, we have for any $\epsilon > 0$ and any $\nu \geq 2$ that

$$\mathbb{P}\Big(\big|\widetilde{Q}_h^k(s, a) - \overline{Q}_h^k(s, a)\big| > \nu\epsilon\Big) \leq 2 \exp\Big\{-8 \exp\{-2(C_\beta + \epsilon)^2\} \cdot N_k \epsilon^2 / \pi\Big\} + 2 \exp\{-2 N_k \epsilon^2\}.$$

*Proof.* For notational simplicity, we omit the state-action pair $(s, a)$, which is fixed throughout this proof. By Lemma E.1, we have

$$\mathbb{P}\big(\widetilde{Q}_h^k > \overline{Q}_h^k + \nu\epsilon\big) \tag{C.1}$$

$$= \mathbb{P}\big((1 - \nu) \cdot \widehat{Q}_h^k + \nu \cdot Q_h^{k,(n_k)} > \overline{Q}_h^k + \nu\epsilon\big)$$

$$= \mathbb{P}\Big((1 - \nu) \cdot \big(\widehat{Q}_h^k - \langle\phi, \overline{\omega}_h^k\rangle\big) + \big[\nu \cdot \big(Q_h^{k,(n_k)} - \langle\phi, \overline{\omega}_h^k\rangle\big) - C_k \cdot \nu\sigma \cdot \sqrt{\phi^\top (\Lambda_h^k)^{-1}\phi}\big] > \nu\epsilon\Big)$$

$$\leq \underbrace{\mathbb{P}\big((1 - \nu) \cdot \big(\widehat{Q}_h^k - \langle\phi, \overline{\omega}_h^k\rangle\big) > \nu\epsilon\big)}_{= \, p_1} + \underbrace{\mathbb{P}\Big(\big(Q_h^{k,(n_k)} - \langle\phi, \overline{\omega}_h^k\rangle\big) - C_k\sigma \cdot \sqrt{\phi^\top (\Lambda_h^k)^{-1}\phi} > \epsilon\Big)}_{= \, p_2}.$$

Before upper bounding $p_1$ and $p_2$, we charatcerize $\langle\phi, \omega_h^{k,i}\rangle$ for $i \in [N_k]$ in the following. By the proof of Lemma E.1, since the state-action pair $(s, a) \in \mathcal{S} \times \mathcal{A}$ is fixed, $\langle\phi, \omega_h^{k,i}\rangle$ is a one-dimensional Gaussian with mean $\langle\phi, \overline{\omega}_h^k\rangle$ and variance $\sigma^2 \cdot \phi^\top (\Lambda_h^k)^{-1}\phi$. Also, by definition of $\Lambda_h^k$ in (5.5), we have $\Lambda_h^k \succeq \sigma^2 \cdot I_d$, which, together with the assumption that $\|\phi\| \leq 1$, gives

$$\sigma^2 \cdot \phi^\top (\Lambda_h^k)^{-1}\phi \leq \big\|(\Lambda_h^k)^{-1}\big\| \cdot \|\phi\|^2 \leq \sigma^2 \cdot (1/\sigma^2) \cdot 1 = 1. \tag{C.2}$$

**Bounding $p_1$:** For $\nu \geq 2$, we have

$$p_1 = \mathbb{P}\Big(\widehat{Q}_h^k < \langle\phi, \overline{\omega}_h^k\rangle - [\nu/(\nu - 1)] \cdot \epsilon\Big)$$

$$\leq \mathbb{P}\big(\widehat{Q}_h^k < \langle\phi, \overline{\omega}_h^k\rangle - 2\epsilon\big) = \mathbb{P}\Big(\frac{1}{N_k} \sum_{i=1}^{N_k} \langle\phi, \omega_h^{k,i}\rangle < \langle\phi, \overline{\omega}_h^k\rangle - 2\epsilon\Big).$$

Since $\langle\phi, \omega_h^{k,i}\rangle$ for $i \in [N_k]$ are one-dimensional Gaussian with mean $\langle\phi, \overline{\omega}_h^k\rangle$ and variance $\sigma^2 \cdot \phi^\top (\Lambda_h^k)^{-1}\phi$, by Hoeffding's inequality we further obtain

$$p_1 \leq \exp\Big\{-\frac{4\epsilon^2}{2 N_k^{-1} \cdot \sigma^2 \cdot \phi^\top (\Lambda_h^k)^{-1}\phi}\Big\} \leq \exp\{-2 N_k \epsilon^2\}, \tag{C.3}$$

where the second inequality follows from (C.2).

**Bounding $p_2$:** Let $\overline{Q}_h^{k,(n_k)} = \min\{\langle\phi, \overline{\omega}_h^k\rangle + C_k\sigma \cdot \sqrt{\phi^\top (\Lambda_h^k)^{-1}\phi}, H - h + 1\}$. Note that

$$\Big\{\big(Q_h^{k,(n_k)} - \langle\phi, \overline{\omega}_h^k\rangle\big) - C_k\sigma \cdot \sqrt{\phi^\top (\Lambda_h^k)^{-1}\phi} > \epsilon\Big\} = \Big\{\frac{1}{N_k} \sum_{i=1}^{N_k} \mathbb{1}\{Q_h^{k,i} > \overline{Q}_h^{k,(n_k)} + \epsilon\} > \frac{n_k}{N_k}\Big\},$$

where the indicators $\mathbb{1}\{Q_h^{k,i} > \overline{Q}_h^{k,(n_k)} + \epsilon\}$ for $i \in [N_k]$ are independently identical distributed random variables with expectation

$$\mathbb{E}\big[\mathbb{1}\{Q_h^{k,i} > \overline{Q}_h^{k,(n_k)} + \epsilon\}\big] = \mathbb{P}\big(Q_h^{k,i} > \overline{Q}_h^k + \epsilon\big) = n_k / N_k - \delta_{h,+\epsilon}^k. \tag{C.4}$$

Here

$$\delta_{h,+\epsilon}^k = \mathbb{P}(Q_h^{k,1} > \overline{Q}_h^{k,(n_k)}) - \mathbb{P}(Q_h^{k,1} > \overline{Q}_h^{k,(n_k)} + \epsilon).$$

Combining (C.1) and (C.4) we have

$$\mathbb{P}(\widetilde{Q}_h^k > \overline{Q}_h^{k,(n_k)} + \epsilon) = \mathbb{P}\left(\frac{1}{N_k}\sum_{i=1}^{N_k} \mathbb{1}\{Q_h^{k,i} > \overline{Q}_h^{k,(n_k)} + \epsilon\} > \mathbb{E}\left[\mathbb{1}\{Q_h^{k,1} > \overline{Q}_h^{k,(n_k)} + \epsilon\}\right] + \delta_{h,+\epsilon}^k\right).$$
(C.5)

Since the indicator is bounded within $[0,1]$, by Hoeffding's inequality we obtain

$$\mathbb{P}(\widetilde{Q}_h^k > \overline{Q}_h^{k,(n_k)} + \epsilon) \le \exp(-2N_k \cdot (\delta_{h,\epsilon}^k)^2).$$
(C.6)

It ramains to characterize $\delta_{h,\epsilon}^k$. Thus, we have

$$\delta_{h,\epsilon}^k = \Phi(C_k) - \Phi\left(C_k + \frac{\epsilon}{\sqrt{\sigma^2 \cdot \phi^\top (\Lambda_h^k)^{-1}\phi}}\right) \ge \frac{2\exp\{-(C_\beta + \epsilon)^2\}}{\sqrt{\pi}} \cdot \epsilon,$$
(C.7)

which is a consequence of $\mathrm{d}(\Phi(z))/\mathrm{d}z = (2/\sqrt{\pi}) \cdot \exp(-z^2)$ and $C_k \le C_\beta$. Taking (C.7) into (C.6), we obtain

$$p_2 \le \exp\left\{-8\exp\{-2(C_\beta + \epsilon)^2\} \cdot N_k\epsilon^2/\pi\right\}.$$
(C.8)

Taking (C.3) and (C.8) into (C.1), we have

$$\mathbb{P}(\widetilde{Q}_h^k > \overline{Q}_h^k + \nu\epsilon) \le \exp\left\{-8\exp\{-2(C_\beta + \epsilon)^2\} \cdot N_k\epsilon^2/\pi\right\} + \exp\{-2N_k\epsilon^2\}.$$
(C.9)

Similarly, we also have

$$\mathbb{P}(\widetilde{Q}_h^k < \overline{Q}_h^k - \nu\epsilon) \le \exp\left\{-8\exp\{-2(C_\beta + \epsilon)^2\} \cdot N_k\epsilon^2/\pi\right\} + \exp\{-2N_k\epsilon^2\},$$

combining which with (C.9) concludes the proof. $\qquad\square$

Next, we introduce three useful lemmas to connect the above concentration lemma for a fixed state-action pair to the desired uniform concentration guarantee over all $(s,a) \in \mathcal{S} \times \mathcal{A}$.

**Lemma C.2** (Proposition 1 in [20]). Let $v$ be a Gaussian random vector in $\mathbb{R}^d$ with mean zero and covariance matrix $\Sigma \in \mathbb{R}^{d \times d}$. For any $t > 0$, it holds that

$$\mathbb{P}(\|v\|^2 > \mathrm{tr}(\Sigma) + 2\sqrt{\mathrm{tr}(\Sigma^2) \cdot t} + 2\|\Sigma\| \cdot t) \le \exp(-t).$$

Let $v_h^{k,i} = \omega_h^{k,i} - \overline{\omega}_h^k \in \mathbb{R}^d$ for $i \in [N]$. We have $v_h^{k,i} \overset{\text{i.i.d.}}{\sim} \mathcal{N}(0, \sigma^2 \cdot (\Lambda_h^k)^{-1})$ for $i \in [N]$. By Lemma C.2, we have with probability at least $1 - \exp(-t)$ that

$$\|v_h^{k,i}\|^2 \le \mathrm{tr}((\Lambda_h^k)^{-1}) + 2\sqrt{\mathrm{tr}((\Lambda_h^k)^{-2}) \cdot t} + 2\|(\Lambda_h^k)^{-1}\| \cdot t$$
$$\le d/\sigma^2 + 2\sqrt{dt}/\sigma^2 + 2t/\sigma^2.$$
(C.10)

Combining Lemma E.2 and (C.10), we further have with probability at least $1 - \exp(t)$ that

$$\|\omega_h^{k,i}\|^2 = \|\overline{\omega}_h^k + v_h^{k,i}\|^2 \le 2\|\overline{\omega}_h^k\|^2 + 2\|v_h^{k,i}\|^2 \le (8dH^2k + 2d + 4\sqrt{dt} + 4t)/\sigma^2$$
$$\le \underbrace{(10dH^2k + 4\sqrt{dt} + 4t)/\sigma^2}_{W(t)^2}.$$
(C.11)

**Lemma C.3.** For $\{w^i\}_{i \in [N]} \subset \mathbb{R}^d$ such that $\|w^i\| \le W$ for all $i \in [N]$ and vectors $\phi, \phi' \in \mathbb{R}^d$ such that $\|\phi - \phi'\| \le \varepsilon$, let $\widetilde{Q}$ and $\widetilde{Q}'$ be the $m$-th ($1 \le m \le N$) order statistics in $\{\langle\phi, w^i\rangle\}_{i \in [N]}$ and $\{\langle\phi', w^i\rangle\}_{i \in [N]}$ respectively. Then it holds that $|\widetilde{Q} - \widetilde{Q}'| \le \varepsilon W$.

*Proof.* By definition of $\widetilde{Q}$, there exists an index set $I \subset [N]$ such that $|I| = m$ and $\langle \phi, \omega^i \rangle \geq \widetilde{Q}$ for any $i \in I$ and $\langle \phi, \omega^i \rangle \leq \widetilde{Q}$ for any $i \notin I$. Since $\|\phi - \phi'\| \leq \varepsilon$ and $\|w^i\| \leq W$ for all $i \in [N]$, by Hölder's inequality it holds that $\langle \phi', w^i \rangle \geq \widetilde{Q}' - \varepsilon W$ for all $i \in I$, which implies $\widetilde{Q}' \geq \widetilde{Q} - \varepsilon W$. Similarly we also have $\widetilde{Q}' \leq \widetilde{Q} + \varepsilon W$ and thus we finish the proof. $\square$

**Lemma C.4.** For any $\phi, \phi' \in \mathbb{R}^d$ such that $\|\phi - \phi'\| \leq \varepsilon$, let $\overline{Q}_q, \overline{Q}'_q$ be the $q$-quantiles of the random variables $\langle \phi, \omega \rangle$ and $\langle \phi', \omega \rangle$ respectively, where the random vector $\omega$ follows distribution $N(\overline{\omega}, \Sigma)$ with $\|\overline{\omega}\| \leq 2H\sqrt{dk}/\sigma$ and $\Sigma \preceq I_d$. Then it holds that $|\overline{Q}_q - \overline{Q}'_q| \leq \varepsilon \cdot (2H\sqrt{dk}/\sigma + C_q)$.

*Proof.* Recall that $C_q = \Phi^{-1}(q)$. It holds that

$$\overline{Q}_q = \langle \phi, \overline{\omega} \rangle + C_q \cdot \sqrt{\phi^\top \Sigma \phi}, \quad \overline{Q}'_q = \langle \phi', \overline{\omega} \rangle + C_q \cdot \sqrt{(\phi')^\top \Sigma \phi'}.$$

Thus we have

$$\begin{aligned}
|\overline{Q}_q - \overline{Q}'_q| &= \left| \langle \phi - \phi', \overline{\omega} \rangle + c_q \cdot \left( \sqrt{\phi^\top \Sigma \phi} - \sqrt{(\phi')^\top \Sigma \phi'} \right) \right| \\
&\leq \|\phi - \phi'\| \cdot \|\overline{\omega}\| + C_q \cdot \|A(\phi - \phi')\| \\
&\leq \varepsilon \cdot \left( \|\overline{\omega}\| + C_q \cdot \|A\| \right) \leq \varepsilon \cdot \left( 2H\sqrt{dk}/\sigma + C_q \right),
\end{aligned}$$

where $\Sigma = A^\top A$, and the last inequality follows from $\|A\| = \|\Sigma\|^{1/2} \leq 1$. $\square$

Now we are ready to prove Lemma 5.1.

*Proof of Lemma 5.1.* Let $N_\varepsilon(\mathcal{B}) = (1 + 2/\varepsilon)^d$ be the $\varepsilon$-covering number of the unit ball $\mathcal{B}$ in $\mathbb{R}^d$. For any $(s, a) \in \mathcal{S} \times \mathcal{A}$, since $\|\phi(s, a)\| \leq 1$, there exists a vector $\phi' \in \mathbb{R}^d$ in the $\varepsilon$-covering such that

$$\|\phi(s, a) - \phi'\| \leq \varepsilon.$$

We denote by $\widetilde{Q}_h^{k'}$ and $\overline{Q}_h^{k'}$ the empirical and true $(n_k/N_k)$-quantile of $\langle \phi', \omega_h^{k,1} \rangle$ respectively. Then we have the following decomposition

$$\left| \widetilde{Q}_h^k(s, a) - \overline{Q}_h^k(s, a) \right| \leq \left| \widetilde{Q}_h^k(s, a) - \widetilde{Q}_h^{k'} \right| + \left| \widetilde{Q}_h^{k'} - \overline{Q}_h^{k'} \right| + \left| \overline{Q}_h^{k'} - \overline{Q}_h^k(s, a) \right|. \tag{C.12}$$

Plugging Lemmas C.1, C.3 and C.4 into (C.12), and recalling $W(t)$ defined in (C.11) and using $\sigma = 1$, we have

$$\left| \widetilde{Q}_h^k(s, a) - \overline{Q}_h^k(s, a) \right| \leq \varepsilon \cdot W(t) + \nu\epsilon + \varepsilon \cdot \left( 2H\sqrt{dk} + C_k \right), \tag{C.13}$$

for all $(x, a, k, h) \in \mathcal{S} \times \mathcal{A} \times [K] \times [H]$, $p \in (0, 1]$ and $t, \epsilon, \varepsilon > 0$, with probability at least

$$1 - 2N_\varepsilon(\mathcal{B})T \cdot \left( \exp\left\{ -8\exp\{-2(C_\beta + \epsilon)^2\} \cdot N_k \epsilon^2/\pi \right\} + \exp\{-2N_k\epsilon^2\} \right) - NT\exp(-t). \tag{C.14}$$

In what remains, we take $\epsilon = \sqrt{\iota/(dk)}$ and $\varepsilon = 1/(5dk)$. First, we have $(C_\beta + \epsilon)^2 \leq 2C_\beta^2 + 2\epsilon^2 = 2c_\beta^2 \cdot \iota + 2\iota/dk \leq 2(c_\beta^2 + 1) \cdot \iota$, taking which into (C.14) and recalling that $\iota = \log(3dT/p)$, we obtain

$$8\exp\{-2(C_\beta + \epsilon)^2\}/\pi \geq 8\exp\{-4(c_\beta^2 + 1)\} \cdot (3dT/p)^{-4}/\pi = (1/c'_\beta) \cdot (dT/p)^{-4},$$

where $c'_\beta = 81\pi \exp\{4(c_\beta^2 + 1)\}/8$. Thus, if

$$N_k = 2c'_\beta \cdot d^6 T^4 k/p^4 \geq \frac{\log\big(18N_\varepsilon(\mathcal{B}) \cdot T/p\big)}{8\exp\{-2(C_\beta + \epsilon)^2\} \cdot \epsilon^2/\pi}, \tag{C.15}$$

and

$$t = \log\big(9N_k T/p\big), \tag{C.16}$$

we have the probability in (C.14) being at least $1 - p/3$. Taking (C.15) into (C.16), we further have

$$\sqrt{t} = \sqrt{\iota + \log(3N_k/d)} \leq \sqrt{6\iota + 4(1 + c_\beta^2)} \leq \sqrt{6\iota} + 2 + 2c_\beta. \tag{C.17}$$

Next, taking (C.17) into (C.13), we obtain

$$W(t) = \left(10dH^2k + 4\sqrt{dt} + 4t\right)^{1/2} \leq (7 + 4c_\beta) \cdot H\sqrt{dk\iota}. \tag{C.18}$$

Finally, using $C_k \leq C_\beta = c_\beta \cdot \sqrt{\iota}$ and (C.18), we know that with probability at least $1 - p/3$,

$$\begin{aligned}
\left|\widetilde{Q}_h^k(s, a) - \overline{Q}_h^k(s, a)\right| &\leq \varepsilon \cdot \left(W(t) + 2H\sqrt{dk} + C_\beta\right) + \nu\epsilon \\
&= \frac{1}{5dk} \cdot \left((7 + 4c_\beta) \cdot H\sqrt{dk\iota} + 2H\sqrt{dk}\right) + \frac{c_\beta \cdot \sqrt{\iota}}{5dk} + H\sqrt{d\iota/k} \\
&\leq \left(c_\beta/\sqrt{d} + 3\right) \cdot H\sqrt{d\iota/k},
\end{aligned}$$

which concludes the proof. $\qquad\square$

# D  Proof of Lemma 5.2

**Lemma D.1** (Lemma D.4 in [24]). *Let $\{s^\tau\}_{\tau \geq 0}$ be a stochastic process in the state space $\mathcal{S}$ with corresponding filtration $\{\mathcal{F}^\tau\}_{\tau \geq 0}$. Let $\{\phi^\tau\}_{\tau \geq 0} \subset \mathbb{R}^d$ be a stochastic process where $\phi^\tau \in \mathcal{F}^{\tau-1}$ and $\|\phi^\tau\| \leq 1$ for all $\tau \geq 0$. Let $\Lambda^k = \sum_{\tau=1}^{k} \phi^\tau(\phi^\tau)^\top + \sigma^2 \cdot I_d$. Then for any $\delta > 0$, with probability at least $1 - p$, for all $k \geq 0$ and any $V \in \mathcal{V}$ so that $\sup_s |V(s)| \leq H$, we have*

$$\left\|\sum_{\tau=1}^{k} \phi^\tau \cdot \left(V(s^\tau) - \mathbb{E}[V(s^\tau) \mid \mathcal{F}^{\tau-1}]\right)\right\|_{(\Lambda^k)^{-1}}^2 \leq 4H^2 \cdot \left[\frac{d}{2} \cdot \log\left(\frac{k + \sigma^2}{\sigma^2}\right) + \log\frac{N_\varepsilon(\mathcal{V})}{p}\right] + \frac{8k^2\varepsilon^2}{\sigma^2},$$

*where $N_\varepsilon(\mathcal{V})$ is the $\varepsilon$-covering number of the function class $\mathcal{V}$ with respect to the distance $d(V, V') = \sup_{s \in \mathcal{S}} |V(s) - V'(s)|$.*

**Lemma D.2** (Lemma D.6 in [24]). *Let $\mathcal{V}_h$ be the class of functions mapping from $\mathcal{S}$ to $\mathbb{R}$ with the form*

$$V_h(\cdot) = \max_{a \in \mathcal{A}}\left\{\min\left\{\langle\phi(\cdot, a), \omega\rangle + C_k \cdot \nu\sigma \cdot \sqrt{\phi(\cdot, a)^\top \Lambda^{-1}\phi(\cdot, a)}, H - h + 1\right\}^+\right\},$$

*where the parameters $(\omega, C_k, \Lambda)$ satisfy $\|\omega\| \leq L$, $C_k \in [C_\alpha, C_\beta]$, and $\lambda_{\min}(\Lambda) \geq \sigma^2$. Assume that $\|\phi(s, a)\| \leq 1$ for all $(s, a) \in \mathcal{S} \times \mathcal{A}$, and let $N_\varepsilon(\mathcal{V}_h)$ be the $\varepsilon$-covering number of $\mathcal{V}$ with respect to the distance $d(V_h, V_h') = \sup_{s \in \mathcal{S}} |V_h(s) - V_h'(s)|$. Then,*

$$\log N_\varepsilon(\mathcal{V}_h) \leq d \cdot \log\left(1 + \frac{4L}{\varepsilon}\right) + d^2 \cdot \log\left(1 + \frac{8C_\beta^2 \cdot \nu^2\sqrt{d}}{\varepsilon^2}\right).$$

*Proof of Lemma 5.2.* First, recall that we defined in (B.4) and (B.5) the filtration $\{\mathcal{F}_h^k\}_{k \in [K], h \in [H]}$, based on which we can define another filtration $\{\mathcal{G}_h^k\}_{k \in [K], h \in [H]}$ by

$$\mathcal{G}_h^{k-1} = \mathcal{F}_h^k \cup \{a_h^k\}, \quad (k, h) \in [K] \times [H].$$

Then, by definition of $[\mathcal{P}_h V]$, we have for all $\tau \in [k-1]$ that

$$\begin{aligned}
[\mathcal{P}_h \overline{V}_{h+1}^k](s_h^\tau, a_h^\tau) &= \mathbb{E}_{s_{h+1} \sim \mathcal{P}_h(\cdot \mid s_h^\tau, a_h^\tau)}\left[\overline{V}_{h+1}^k(s_{h+1}) \mid s_h^\tau, a_h^\tau\right] \\
&= \mathbb{E}_{s_{h+1} \sim \mathcal{P}_h(\cdot \mid s_h^\tau, a_h^\tau)}\left[\overline{V}_{h+1}^k(s_{h+1}) \mid \mathcal{G}_h^{\tau-1}\right].
\end{aligned}$$

Thus, by Lemmas D.1, D.2 and E.2, we have with probability at least $1 - p/3$ that

$$\left\|\sum_{\tau=1}^{k-1} \phi_h^\tau \cdot \left(\overline{V}_{h+1}^k(s_{h+1}^\tau) - [\mathcal{P}_h \overline{V}_{h+1}^k](s_h^\tau, a_h^\tau)\right)\right\|_{(\Lambda_h^k)^{-1}}^2 \tag{D.1}$$

$$\leq 4H^2 \cdot \left[\frac{d}{2} \cdot \log\left(\frac{k + \sigma^2}{\sigma^2}\right) + \log\left(\frac{3}{p}\right) + d \cdot \log\left(1 + \frac{8H\sqrt{dk}}{\varepsilon\sigma}\right) + d^2 \cdot \log\left(1 + \frac{8C_\beta^2 \cdot \nu^2\sqrt{d}}{\varepsilon^2}\right)\right] + \frac{8k^2\varepsilon^2}{\sigma^2}.$$

In (D.1), taking $\sigma = 1$, $\varepsilon = dH/k$, $\nu = dH$ and $C_\beta = c_\beta \cdot \sqrt{\iota}$, we obtain that there exists an absolute constant $C > 0$ such that

$$\left\| \sum_{\tau=1}^{k-1} \phi_h^\tau \cdot \left( \overline{V}_{h+1}^k(s_{h+1}^\tau) - [\mathcal{P}_h \overline{V}_{h+1}^k](s_h^\tau, a_h^\tau) \right) \right\|_{(\Lambda_h^k)^{-1}}^2 \leq C^2 \cdot d^2 H^2 \chi,$$

for all $(k, h) \in [K] \times [H]$, where $\chi = \log[3(1 + c_\beta)dT/p]$. Therefore, we finish the proof. $\qquad\square$

# E  Supporting Lemmas

**Lemma E.1.** For $\overline{Q}_h^k(s, a)$ defined in (5.2), we have for all $(s, a) \in \mathcal{S} \times \mathcal{A}$ that

$$\overline{Q}_h^k(s, a) = \min\left\{ \langle \phi(s, a), \overline{\omega}_h^k \rangle + C_k \cdot \nu\sigma \cdot \sqrt{\phi(s, a)^\top (\Lambda_h^k)^{-1} \phi(s, a)}, H - h + 1 \right\},$$

where $\overline{\omega}_h^k = (\Lambda_h^k)^{-1} \sum_{\tau=1}^{k-1} y_h^\tau \cdot \phi_h^\tau$ is defined in (5.4).

*Proof.* When the prior and the likelihood are Gaussian, the posterior of $\omega$ is Gaussian distribution with mean

$$\overline{\omega}_h^k = \mathbb{E}\big[\omega \,\big|\, \{\widetilde{V}_{h+1}^k(s_{h+1}^\tau)\}_{\tau \in [k-1]}, \mathcal{D}_h^k\big]$$

$$= \underset{\omega \in \mathbb{R}^d}{\operatorname{argmax}} \left\{ -\frac{1}{2} \cdot \|\omega\|^2 - \frac{1}{\sigma^2} \sum_{\tau=1}^{k-1} \big( y_h^\tau - Q(s_h^\tau, s_h^\tau; \omega) \big)^2 \right\} = (\Lambda_h^k)^{-1} \sum_{\tau=1}^{k-1} y_h^\tau \cdot \phi_h^\tau,$$

where $\Lambda_h^k$ is defined in (5.5), and $y_h^\tau$ is defined in (3.2). Also, we have the covariance of the posterior as

$$\operatorname{Cov}\Big( \omega \,\Big|\, \{\widetilde{V}_{h+1}^k(s_{h+1}^\tau)\}_{\tau \in [k-1]}, \mathcal{D}_h \Big) = \sigma^2 \cdot (\Lambda_h^k)^{-1},$$

from which we obtain for all $i \in [N_k]$ that

$$\langle \phi(s, a), \omega_h^{k,i} \rangle \sim \mathcal{N}\Big( \langle \phi(s, a), \overline{\omega}_h^k \rangle, \sigma^2 \cdot \phi(s, a)^\top (\Lambda_h^k)^{-1} \phi(s, a) \Big).$$

Thus, we have

$$\mathbb{E}_\omega \big[ (1 - \nu) \cdot \widehat{Q}_h^k(s, a) + \nu \cdot Q_h^{k,(n_k)}(s, a) \big]$$

$$= (1 - \nu) \cdot \langle \phi(s, a), \overline{\omega}_h^k \rangle + \nu \cdot \Big( \langle \phi(s, a), \overline{\omega}_h^k \rangle + \Phi^{-1}(n_k/N_k) \cdot \sqrt{\sigma^2 \cdot \phi(s, a)^\top (\Lambda_h^k)^{-1} \phi(s, a)} \Big)$$

$$= \langle \phi(s, a), \overline{\omega}_h^k \rangle + C_k \cdot \nu\sigma \cdot \sqrt{\phi(s, a)^\top (\Lambda_h^k)^{-1} \phi(s, a)},$$

plugging which into definition of $\overline{Q}_h^k$ in (5.2) concludes the proof. $\qquad\square$

**Lemma E.2** (Lemma B.2 in [24]). For $\overline{\omega}_h^k$ defined in Lemma (E.1) and all $(k, h) \in [K] \times [H]$, we have

$$\|\overline{\omega}_h^k\| \leq 2H\sqrt{dk}/\sigma.$$

**Lemma E.3** (Lemma D.1 in [24]). Let $\Lambda^k = \sum_{\tau=1}^k \phi^\tau (\phi^\tau)^\top + \lambda \cdot I_d$, where $\phi^\tau \in \mathbb{R}^d$ and $\lambda > 0$. Then

$$\sum_{\tau=1}^k (\phi^\tau)^\top (\Lambda^k)^{-1} \phi^\tau \leq d.$$

**Lemma E.4** (Elliptical Potential Lemma in, e.g., [10, 24]). Let $\{\phi^\tau\}_{\tau \geq 0} \subset \mathbb{R}^d$ be a sequence satisfying $\sup_{\tau \geq 0} \|\phi^\tau\| \leq 1$. Let $\Lambda^0 \in \mathbb{R}^{d \times d}$ be a positive definite matrix. For any $k \geq 0$, we define $\Lambda^k = \Lambda^0 + \sum_{\tau=1}^k \phi^\tau (\phi^\tau)^\top$. Then, if $\lambda_{\min}(\Lambda^0) \geq 1$, we have

$$\log\left( \frac{\det(\Lambda^k)}{\det(\Lambda^0)} \right) \leq \sum_{\tau=1}^k (\phi^\tau)^\top (\Lambda^{\tau-1})^{-1} \phi^\tau \leq 2 \log\left( \frac{\det(\Lambda^k)}{\det(\Lambda^0)} \right).$$

# F Illustrative Experiments

## F.1 Synthetic Environment

In this section, we perform illustrative experiments on a synthetic dataset. We compare BooVI with three baseline algorithms: Random-Exploration, Epsilon-Greedy [47], and LSVI-UCB [24].

**Environment.** The construction of the synthetic MDP environment with simplex feature follows that of [58]. The constructed MDP has $|\mathcal{S}| = 15$, $|\mathcal{A}| = 7$, $H = 10$, $d = 10$, and $K = 300$. In the constructed MDP, there is only one good chain that leads to a huge reward at (only) the end of the episode. Otherwise, the agent receives small positive rewards in the suboptimal chains. Such an MDP requires the agent to perform deep exploration [30] in order to reach near-optimal policy rather than being attracted to the small suboptimal rewards.

**Baseline.** The Random-Exploration baseline takes actions uniformly randomly throughout all the episodes. The Epsilon-Greedy baseline takes the greedy action according to the current estimates of $\{Q_h\}_{h \in [H]}$ with probability of $1 - \epsilon$, and takes the uniformly random action with probability $\epsilon$. Here we set $\epsilon = 0.05$. In LSVI-UCB, we tune the parameter $\beta$ for a best outcome. In the reported results, we use $\beta = 0.7$.

**BooVI Setup.** Due to the linear structure in the synthetic MDP, we are able to compute the matrix inverse $(\Lambda_h^k)^{-1}$. Thus, we perform posterior weight sampling via directly sampling from the exact posterior distribution $\mathcal{N}(\omega_h^k, (\Lambda_h^k)^{-1})$. In the reported results, we use the extrapolation parameter $\nu = 1.4$, the order parameter $n_k/N_k = 0.6$, and three different of posterior sample sizes $N_k \equiv 5$, $N_k \equiv 10$, and $N_k \equiv 100$.

In addition, to test the case with approximate posterior sampling, we perform BooVI-Langevin, which uses Langevin dynamics in (3.4) to sample posterior weights. In BooVI-Langevin, we use the same $\nu$ and $n_k$ as BooVI, and we use $N_k \equiv 10$. For the Langevin dynamics sampling part, we choose the starting weight as $\omega_h^k$ and use the stepsize $\eta_t = 0.01/t$. To avoid getting too correlated posterior weights, we use 5 Langevin iterates to warm up, which are thrown away, and keep 1 posterior weight every 3 iterates after the warm-up phase.

**Result.** In the following, we report the cumulative rewards for BooVI, BooVI-Langevin, and the three baselines. The error bars reflect one standard deviation with 10 trials.

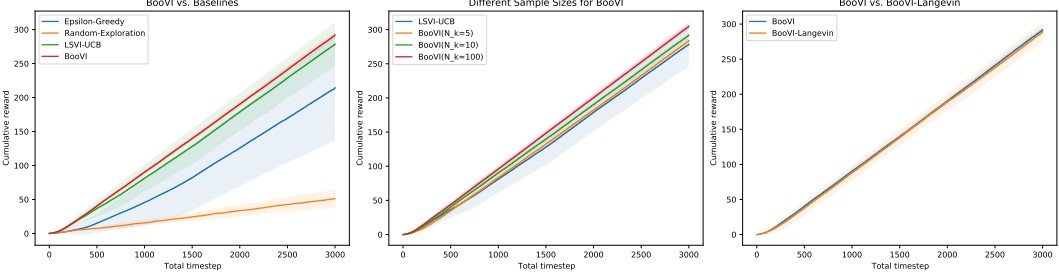

Left: Comparison between BooVI with $N_k \equiv 10$ and the baselines.
Middle: Comparison for different sample sizes $N_k = 5, 10, 100$ with LSVI-UCB as the baseline.
Right: Comparison between BooVI with $N_k = 10$ and BooVI-Langevin with $N_k = 10$.

In the left figure, we can observe that BooVI attains slightly better performance as LSVI-UCB (both outperform Epsilon-Greedy and Random-Exploration) but has much smaller standard deviation on the performance among trials. The middle figure shows that the larger $N_k$ only improves the performance of BooVI marginally. The right figure shows that BooVI and BooVI-Langevin also has similar performance. However, BooVI-Langevin has larger standard deviation on the performance due to the extra stochasticity brought by the Langevin dynamics.

## F.2 Freeway Game

We present the following illustrative result on the freeway game in Atari from OpenAI gym [7]. Here we choose $N_k \equiv 5$ posterior sample size and $\nu = 50$ extrapolation parameter throughout the training process. The standard deviation is plotted using $5$ trials.

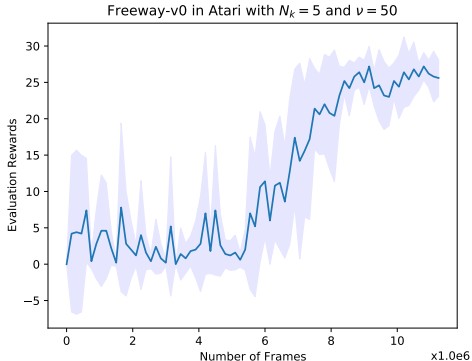

Freeway_v0: Scores achieved by BooVI with $N_k \equiv 5$ and $\nu = 50$.

Without any modification, BooVI achieves final performances ranging from $25$ to $30$ in freeway. Considering that only $N_k \equiv 5$ is used, the computational overhead is much less than what' suggested by the theory.

## F.3 Discussion

As we see in the experimental results in synthetic environment, BooVI and BooVI-Langevin have comparable performance compared to LSVI-UCB at the cost of extra memory allocation for $N_k$ copies of $Q$-function parameters and extra computational cost for evaluating the sampled $Q$-functions for $N_k$ times. However, the requirements on the extrapolation parameter $\nu$, and the sample size $N_k$ are much milder than the requirements in the theoretical analysis, which results in milder computational overhead and memory allocation compared to LSVI/LSVI-UCB. We speculate that this is due to the pessimism nature of the analysis for establishing upper bounds. Moreover, in the comparison between BooVI and BooVI-Langevin, we see similar performance without using large number of Langevin iterates (35 total iterates). This suggests that BooVI is robust to approximate posterior sampling.

The aforementioned phenomenons in the experiment results for the synthetic MDP are encouraging. It would be interesting to see the performance of BooVI in more challenging environments, which we leave to our future work.