# OpenReview forum: "BooVI: Provably Efficient Bootstrapped Value Iteration"
_NeurIPS.cc/2021/Conference — NeurIPS 2021 Poster_

### Official Review · Reviewer_GkJA · 2021-07-13

**Rating:** 6
**Confidence:** 4

**Summary:**

This paper proposes a posterior sampling-based algorithm BooVI, which suffices for general function approximation purposes. In the case of linear MDP setting, it achieves $\tilde{O}\sqrt{d^3H^3T}$ regret with Gaussian posterior sampling.

**Ethical Concerns:**

No.

**Limitations And Societal Impact:**

Yes.

**Main Review:**

Overall, this paper provides an interesting posterior sampling algorithm BooVI that works for general MDPs. The technique and proofs look sound to me. However, I have the following two questions.

1. The regret obtained for the linear MDP has no improvement over Jin et. al, is that due to the limitation of the technical tools or due to the algorithmic design?

2. The paper would be more complete if you can instantiate more examples, e.g. linear mixture MDP. Could you provide an explanation of what the theoretical result would look like for your algorithm?

Given that the paper provides some nice contributions, I would like to raise my score if the authors can address my concerns.



**Time Spent Reviewing:**

8 Hours

---

> ### Author Response · Authors · 2021-08-10
> **Response to Reviewer GkJA**
>
> Regret bound: The regret bound matches [Jin et al., 2019] due to the use Hoeffding type of inequalities as technical tools. Our regret bound is near-optimal in a sense that the dependency over $T$ is optimal. The main purpose of this paper is to design a bootstrapping exploration algorithm for RL tasks that is both sample efficient and can handle general function approximation. On top of that, we establish theoretical guarantee of regret for a class of linear MDPs. Sharpening the regret bound, while being of independent interest, falls out of the scope in this work. We will consider this as a future follow up of our work.
>
> More general setting: Although the algorithm design is inspired by linear MDPs, generalizing our results to boarder classes of MDPs are possible. For the more technical challenging linear mixture MDPs, we will need to adjust the feature of Q-function to take the form of $\phi_{V}$ (see e.g., [3]), and potentially corresponding modifications on the posterior distribution (which might take the form of Gaussian mixture) and the requirement on extrapolation parameter $\nu$. The convergence results can possibly stay the same dependency over $T$ (i.e., $\tilde{O}(\sqrt{T})$), which would, however, take significant effort to prove. Also, we can similarly establish Lemma 5.3 for generalized linear model [1], and kernel and over-parameterized NN [2]. Please see our response to Reviewer PuQ1 for the setting of approximately low-rank MDPs [Zanette et al., 2019]. We will add more discussion to the possible extension to more general settings our revision.
>
> [1] Wang, et al. ”Optimism in reinforcement learning with generalized linear function approximation.”
>
> [2] Yang, et al. ”Provably Efficient Reinforcement Learning with Kernel and Neural Function Approximations.
>
> [3] Zhou, et al. "Nearly Minimax Optimal Reinforcement Learning for Linear Mixture Markov Decision Processes"

---

### Official Review · Reviewer_PuQ1 · 2021-07-16

**Rating:** 6
**Confidence:** 3

**Summary:**

This paper proposes a modified version of LSVI based on bootstrapping, called BooVI. Different from the famous Bootstrapped DQN, it shows that BooVI is provably efficient under linear MDP; meanwhile, although not theoretically supported, an implementation of BooVI also admits general function approximation like neural networks.

**Limitations And Societal Impact:**

The authors adequately addressed the limitations of their work in Section 6 by presenting them as potential future works. The potential negative social impact is not addressed, which will generally be all potential malicious usage of any reinforcement learning algorithms.

**Main Review:**

Overall, I consider this paper as a good attempt to theoretically support the power of bootstrapping and posterior sampling in reinforcement learning with function approximation. I also consider the specific usage of bootstrapping and Langevin dynamics as novel. The writing is in general a good flow, although with several potential improvements that will be given in "Suggestions on writing" part.

However, I have several concerns about its significance and technical soundness. In particular, Although one of the advantage of BooVI over other provably efficient algorithms is that it practically admits general function approximation, the provided experiments do not contain this part. Other concerns will be given in "Question" part.

Questions:
- How many steps does the Langevin dynamics take?
- The given worst-case regret bound is not strictly an improvement over [1] beucause [1] considers a slightly more general low-rank MDP. Do you think the regret of BooVI is still applicable to the low-rank MDP?
- In the proof of Lemma B.2, why does $\mathcal{F}^1_1$ not contain $\lbrace\omega_h^{1, i}\rbrace$? More importantly, the claim that $\mathbb{E}\left[\xi^k_h(s_h^k, a_h^k, s_{h+1}^k)\mid \mathcal{F}^k_h\right]=0$ makes me feel wired because I think neither $\mathbb{E}\left[\bar{Q}_h^k-Q_h^{\pi^k}(s_h^k, a_h^k)\mid \mathcal{F}_h^k\right]=0$, since $Q^{\pi^k}_h$ is based on the underlying unknown MDP, nor $\mathbb{E}\left[\bar{V}^k_h(s^k_h)-\bar{Q}^k_h(s_h^k, a_h^k)\mid\mathcal{F}_h^k\right]=0$ holds, since $a_h^k$ does not necessarily maximize $\bar{Q}^k_h(s_h^k, \cdot)$. How do the terms in $\xi^k\_h(s_h^k, a\_h^k, s\_{h+1}^k)$ cancel each other under $\mathbb{E}[\cdot\mid\mathcal{F}_h^k]$?

Suggestions on writing:
- Is line 12 in Algorithm 1 necessary? It seems that $\widetilde{Q}^k_h$'s have been generated at line 7 except $\widetilde{Q}^k_1$.
- The writing of proof for Lemma 5.3 may need some improvements because it uses $\bar{\omega}^k_h$ without defining it and implicitly uses Lemma E.1 without mentioning. I apologize if I simply missed them.
- It is probably better to give a slightly more detailed sketch on how the optimism is ensured through Bootstrapping step in Algorithm 2.

```
[1] ZANETTE, A., BRANDFONBRENER, D., PIROTTA, M. and LAZARIC, A. (2019). Frequentist regret bounds for randomized least-squares value iteration. arXiv preprint arXiv:1911.00567.
```

- - -

Score is increased after rebuttal because the technical concerns are well-addressed.



**Time Spent Reviewing:**

8

---

> ### Author Response · Authors · 2021-08-10
> **Response to Reviewer PuQ1**
>
> We thanks the reviewer for the valuable feedback. We appreciate the suggestions on writing and will revise accordingly. We address the major comments in the following.
>
> Langevin dynamics: Theoretically, we need to take Langevin dynamics step polynomial many times for sub-Gaussian tailed posterior to mix (for detailed theory results, please see e.g., [1]). In experiments, using the previous mean of posterior sampling as a hot-start point, we find that one can usually take only very few Lagevin steps (please see Appendix F, where 35 Langevin steps are used).
>
> Low-rank MDP: Our results can be similarly generalized to the low-rank MDP setting. This is due to the fact that approximately low-rank MDP largely preserves the linear structure considered in our paper. Optimism of the bootstrapped Q-function can be preserved with parameters adjusted to adapt the deviations $\Delta_t^r$ and $\Delta_t^P$ and thus ensure efficient exploration. We also expect the regret bound to match that of [Zanette et al., 2019], which has $\tilde{O}(\epsilon T)$ dependency in $T$. Here $\epsilon$ is the upper bound of $\Delta_t^r$ and $\Delta_t^P$. We will add more detailed discussion to this in our revision.
>
> Lemma B.2: Since the filtration already removes the randomness up to $s_h^k$, we have
>
> (i) ${E}[V_h^k(s_h^k) - Q_h^k(s_h^k, a_h^k)\,|\,\mathcal{F}_h^k] = 0$ and  $E[\overline{V}_h^k(s_h^k) - \overline{Q}_h^k(s_h^k, a_h^k)\,|\,\mathcal{F}_h^k] = 0$ by the definitions of $V_h^k$, $Q_h^k$, $\bar{V}_h^k$, and $\bar{Q}_h^k$,
>
> (ii) $E[f(s_{h+1}^k) - \mathcal{P}_h^k f(s_h^k, a_h^k)\,|\,\mathcal{F}_h^k] = 0$ for any function $f$ by the definitions of $\mathcal{P}_h^k$.
>
> Thus the term $\xi(s_h^k, a_h^k, s_{h+1}^k)$ has zero conditional expectation.
>
> Algorithm: Line 12 in Algorithm 1 is necessary since the previous computation only covers $s_h^\tau$ with $\tau$ up to $k-1$, while Line 12 computes bootstrapped Q-functions values for $s_h^k$.
>
> [1] Zhang et al. "A Hitting Time Analysis of Stochastic Gradient Langevin Dynamics"

---

> > ### Comment · Reviewer_PuQ1 · 2021-08-15
> > **Follow-up Question**
> >
> > Thank you very much for the response and my other concerns are will-addressed.
> >
> > However, the proof of Lemma B.2 still is still confusing for me. In particular, why $\mathbb{E}\left[\bar{V}^k_h(s_h^k)-\bar{Q}^k_h(s_h^k, a_h^k)\mid\mathcal{F}\_h^k\right]=0$ holds? **I apologize if I made any mistake.** I understand that the randomness on $s_h^k$ has been removed by $\mathcal{F}\_h^k$. However, the point I'm concerning is that by definition, $a_h^k$ is generated through $a_h^k=\arg\max_{a\in\mathcal{A}}\widetilde{Q}^k_h(s_h^k, a)$ while $\bar{V}\_h^k(s_h^k)=\max_{a\in\mathcal{A}}\bar{Q}^k_h(s_h^k, a)$. Therefore, in order for $\mathbb{E}\left[\bar{V}^k_h(s_h^k)-\bar{Q}^k_h(s_h^k, a_h^k)\mid\mathcal{F}\_h^k\right]=0$ to hold, we need to have $\arg\max_{a\in\mathcal{A}}\bar{Q}^k_h(s_h^k, a)=\arg\max_{a\in\mathcal{A}}\widetilde{Q}^k_h(s_h^k, a)$, which really does not look obvious to me.

---

> > > ### Author Response · Authors · 2021-08-16
> > > **Response to the Follow-up Question: An Easy Fix on a Technical Issue**
> > >
> > > We thank the reviewer for checking our paper in great detail and pointing out this technical issue. We give the following easy fix on it.
> > >
> > > It is true that \\(E[\bar V_h^k(s_h^k) - \bar V_h^k (s_h^k, a_h^k)\, |\, \mathcal{F}_h^k] = 0\\) does not hold for the reason mentioned by the reviewer (and this term is the only one that makes \\(\xi\\) not a martingale). To fix this, we first remove the term \\(\bar V_h^k( s_h^k) - \bar Q_h^k( s_h^k, a_h^k)\\) to make the rest of the terms form a martingale.
> > >
> > > Next, we make the decomposition
> > >
> > > $$
> > > \bar V_h^k( s_h^k) - \bar Q_h^k( s_h^k, a_h^k)  = \bar V_h^k( s_h^k) - \max_a \tilde Q(s_h^k, a) + \max_a \tilde Q(s_h^k, a) - \bar Q_h^k( s_h^k, a_h^k).
> > > $$
> > >
> > > By the definition that \\(\bar V_h^k( s_h^k) = \max_a \bar Q_h^k( s_h^k, a)\\), we have
> > >
> > > $$
> > > \bar V_h^k( s_h^k) - \max_a \tilde Q(s_h^k, a) = \max_a \bar Q_h^k( s_h^k, a) - \max_a \tilde Q(s_h^k, a) \leq \max_a |\bar Q_h^k( s_h^k, a) -  \tilde Q(s_h^k, a)|.
> > > $$
> > >
> > > Since \\(a_h^k\\) is the greedy action with respect to \\(\tilde Q_h^k\\), we have \\(\max_a \tilde Q(s_h^k, a) = \tilde Q(s_h^k, a_h^k)\\), which gives
> > >
> > > $$
> > > \max_a \tilde Q(s_h^k, a) - \bar Q_h^k( s_h^k, a_h^k) = \tilde Q(s_h^k, a_h^k) - \bar Q_h^k( s_h^k, a_h^k).
> > > $$
> > >
> > > Thus, we have
> > >
> > > $$
> > > \bar V_h^k( s_h^k) - \bar Q_h^k( s_h^k, a_h^k)  \leq \max_a |\bar Q_h^k( s_h^k, a) -  \tilde Q(s_h^k, a)| + \tilde Q(s_h^k, a_h^k) - \bar Q_h^k( s_h^k, a_h^k) \leq 2 (c_\beta/\sqrt d + 3) \cdot H \sqrt{d\iota / k},
> > > $$
> > >
> > > with probability at least \\(1 - p/3\\). Here the second inequality follows from Lemma 5.1. Such a term will be summed up over \\(h \in [H]\\) and \\(k \in [K]\\) in our final regret bound (see the first equation in the proof of Lemma 5.5, Line 478). Thus, the extra regret caused by the above arguments is
> > >
> > > $$
> > > \sum_{h = 1}^H \sum_{k = 1}^K \bigl(\bar V_h^k( s_h^k) - \bar Q_h^k( s_h^k, a_h^k)\bigr) \leq 2 (c_\beta/\sqrt d + 3) \cdot H^2 \sqrt{d\iota} \cdot \sum_{k=1}^K 1/\sqrt k = O(\sqrt{d H^3 T \iota}).
> > > $$
> > >
> > > Since our original derived regret bound \\(O(\sqrt{d^3 H^3 T \iota^2})\\) is larger than this extra term, the above fix does not change the regret bound presented in Theorem 4.4.
> > >
> > > We will apply the above fix in our revised version. And we thank the reviewer again for pointing the technical issue out! We hope the above fix satisfactorily addresses the reviewer's concern.

---

> > > > ### Comment · Reviewer_PuQ1 · 2021-08-23
> > > > **Response**
> > > >
> > > > Thank you very much for the response. My concern has been well-addressed and thus I have increased my score.

---

### Official Review · Reviewer_pToa · 2021-07-16

**Rating:** 8
**Confidence:** 3

**Summary:**

The paper studies the low-rank model and proposes a randomized strategy that is competitive in terms of statistical efficiency with the best computationally tractable exploration method for such setting.

**Main Review:**

The work makes very strong contributions to randomized algorithms which are notoriously difficult to study theoretically. This ultimately gives a sqrt{dH} improvement compared to the state of the art randomized RL method (Zanette et al, 2020) with a seemingly clearer and more general analysis.

I am not an expert of Langevin dynamics so I went very fast there. I have a direct question for the authors: is the algorithm computationally tractable, meaning that it requires poly(d,A,H,T,1/\delta) elementary operations? (I understand yes but wanted a direct confirmation).

Unfortunately I did not have the time to carefully examine the proof sketch, but if the result is correct without major caveats then the paper is strong.

I have another question for the authors. The work seems to rely on the linear MDP setting a lot; would the analysis still work in setting of low-inherent Bellman error (e.g., Zanette et al ’20)? I suspect no, because the action value function you maintain is not linear, i.e., it cannot be expressed as \phi(s,a)^\top w.

====

Unchanged after rebuttal

**Time Spent Reviewing:**

1

---

> ### Author Response · Authors · 2021-08-10
> **Response to Reviewer pToa**
>
> We thank the reviewer for the generally positive feedback. We address the questions in the following.
>
> Computational tractability: Yes, BooVI can be implemented with $\mathrm{poly}(d, A, H, 1/\delta)$ many of elementary operations. Specifically, compared to LSVI, the extra computational cost of BooVI comes from posterior sampling (which replaces the process of optimization subproblem for finding a new Q-function and is finished in polynomial time), and $N_k$ times more evaluations (each of which is a $\mathrm{poly}(d, A)$ elementary operation).
>
> More general setting: Establishing theoretical results for BooVI on low inherent Bellman error can be challenging due to the loss of linear representation. Fortunately, low inherent Bellman error setting still preserves approximate linearity. We expect that, with suitable adjustment to posterior distribution (minor adjustment, still Gaussian) and improved technical analysis, our results can be extended to the low inherent Bellman error setting. We expect that the regret bound takes $\tilde{O}(\sqrt{T} + (\mathcal{I} + c)\cdot T)$ dependency in $T$, where $c$ is some constant. However, we are not quite sure whether $\tilde{O}(\sqrt{T})$ can be achieved for BooVI when $\mathcal{I} = 0$ as of now (which is the reason we have additional $c$ after $\mathcal{I}$ in the expected dependency). For other settings, we can similarly establish Lemma 5.3 for generalized linear model [1], and kernel and over- parameterized NN [2]. For the setting of approximately low-rank MDPs [Zanette et al., 2019] and the setting of linear mixture MDPs, please see the response to Reviewer PuQ1 and Reviewer GkJA, respectively. We would like to highlight that, although the theoretical guarantee is only established for a class of linear MDPs, the implementation of the algorithm itself is not limited to linear MDPs (or approximate low-rank MDPs) as opposed to existing works (e.g., [Zanette et al., 2019]).
>
> [1] Wang, et al. ”Optimism in reinforcement learning with generalized linear function approximation.”
>
> [2] Yang, et al. ”Provably Efficient Reinforcement Learning with Kernel and Neural Function Approximations.

---

### Official Review · Reviewer_gbqJ · 2021-07-17

**Rating:** 5
**Confidence:** 4

**Summary:**

The paper proposes an approach for provably efficient reinforcement learning in linear MDPs that avoids constructing explicit confidence intervals. Instead, akin to posterior sampling, this method only requires sampling from a posterior distribution. The paper proves that this approach still yields optimistic value function estimates and retains the regret rate of optimistic LSVI. Because the algorithm requires only  sampling, the paper further claims that the method can be used more easily with general function approximation but does not provide empirical or theoretical results.

**Limitations And Societal Impact:**

The paper does not sufficiently discuss the technical limitations of the proposed approach, concretely the challenges around defining good posteriors for settings beyond linear MDPs.
I do not foresee negative societal impacts of this work.

**Main Review:**

Summary: Overall, I like the boostrap idea but it is not clear how it provides any tangible benefit compared to existing approaches. The paper also has clarity issues.

- Originality: I like the idea of using this concrete form of boostrap for constructing optimistic Q functions and to the best of my knowledge, it has not been proposed or analyzed before. That said, there are existing works that construct Q function estimates by maximizing over multiple posterior samples and which provide theoretical guarantees in the tabular MDP setting.
- Significance: BooVI is likely more conservative than optimistic LSVI, thereby loosing one of the main benefits of RLSVI. This seem to be confirmed by the illustrative experiments in the appendix, where the performance of optimistic LSVI is better or on par than BooVI's. Although the guarantees of BooVI and optimistic LSVI are stronger than that of RLSVI, one would expect it that RLSVI to outperform both without significant parameter tuning.
- Quality: The feature that BooVI only requires posterior sampling and is thus more suitable for use with general function approximation is presented as one of the main advantages. Yet, the paper does not provide any concrete evidence that BooVI is in fact more suitable with general function approximation, neither empirically nor theoretically. This claim is therefore not substantiated. While it is true that sampling is often easier than constructing confidence intervals, this requires a posterior representation that is accurate or which at least induces sufficient exploration. The paper needs to provide concrete examples for this argument to be convincing.
- Clarity: The paper is not sufficiently clear. Here are a few instances that needs improvement:
a) Specifically, the algorithm description in 3 was unnecessary convoluted with cyclic dependencies. For example, the reader does not know where tilde V come from and the reference to Line 1 in Alg 1 seems incorrect. The reader further wonders what the posterior is in line 122 which will only be make clear later. I suggest to first present the posterior given some target values (including the concrete example with Gaussian prior) and then explain how the optimistic target values are generated from the posterior samples, which is the main innovation in this paper.
b) I recommend to restate the condition on C_alpha and C_beta in Thm 4.4 and lemmas as alpha = ... and beta = ... to clarify that they are conditions on those algorithm parameters.
c) The quantity \bar \omega is used extensively in the proof sketch of the main paper and the appendix but never introduced. Lemma E.1 should be stated from and center to clarify this.
d) Generally, it would be more useful for the proof sketch to highlight the differences to existing work, concretely the analysis of optimistic LSVI.


Minor comments and typos:
- Line 168: class MDPs -> class of MDPs
- Alg 1 Ln 7: uising -> using

**Time Spent Reviewing:**

2.5

---

> ### Author Response · Authors · 2021-08-10
> **Response to Reviewer gbqJ**
>
> We thank the reviewer for the valuable feedback. We appreciate the suggestions on presentation improvements and will revise accordingly in the revision. We address the major concerns in the following.
>
> Originality: To the best of our knowledge, BooVI is the first algorithm that can guide exploration under general function approximation, and enjoys the theoretical guarantee in a class of linear MDPs.
>
> Conservativeness: BooVI is not more conservative than optimistic LSVI. Both algorithms are able to attain enough optimism to guide the agent's exploration according to Lemma 5.4 in our paper and the theoretical results in [Jin et al., 2019]. In reality, our experiment results also show that, with tuned parameters, the performance gap between BooVI and optimistic LSVI is almost negligible. Interestingly and surprisingly, we observe that BooVI shows lower performance variance among different trials in the experiment. Lastly, we would like to note that only sufficient optimism is needed for guiding exploration instead of ''more optimistic, the better performance''. This is especially true when truncation is applied to the optimistic Q-function, large optimism would lead to uniform action selection in the early-mid stage of training, which undermines the overall performance.
>
> Parameter tuning: The roles of $\sigma$ in RLSVI, $\beta$ in optimistic LSVI, and $\nu$ and $n_k/N_k$ in BooVI are quite similar. Tuning the parameters in optimistic LSVI is not much harder than in RLSVI. On the other hand, BooVI requires more parameter tuning, but enjoys the benefit of allowing general function approximation (without the need to compute the inverse of potentially huge matrix $\Lambda_h^k$).
>
>
> Function approximation: The implementation of BooVI itself does not involve on any model specific term and thus allows for function approximation. In revision, we will add more experiments using BooVI to guide exploration in more complex environments like Atari to better illustrate its applicability to large problems with function approximation.
>
> Posterior sampling: From a frequentist perspective, using Gaussian likelihood is equivalent to using LSVI. Our analysis partly relies on the sub-Gaussian tail of the posterior. However, BooVI does not specify any particular form of prior/posterior (as long as efficient sampling is possible). The usage of prior is flexible. For example, we can use uninformative prior for generalized linear model or Gaussian process prior for kernel and overparameterized NN. We will add more discussion on the choices of prior/posterior distributions in our revision.

---

> > ### Comment · Reviewer_gbqJ · 2021-09-02
> > **Re: Response to Reviewer gbqJ**
> >
> > I thank the authors for their rebuttal. I am happy that the authors plan to address the clarity issues and add more experiments on using BooVI with more general function approximation.
> > In the paper's current state, the theoretical analysis is limited to linear MDPs and with Gaussian prior and likelihood. I do appreciate that the algorithm's form is amenable to be used with other priors and more general function approximation. However, whether this yields a practical algorithm or whether strong theoretical guarantees are obtainable in such settings (e.g. for low Bellman rank or Bellman-Eluder dimension setting) is not clear. The more extensive experimental evaluation planned by the authors will be one way to show the former but would currently require a leap of faith that those yield reasonable empirical performance.
> >
> > Leaving the potential applicability beyond linear MDPs aside and only considering the presented results for linear MDPs, I am still not sure I see significant benefits over optimistic LSVI or RLSVI. Essentially, the analysis relies on an extremely large number of samples (order $T^4d^6$ from the posterior to show that the mean boostrapped value function is optimistic with high probability. If one were to run this algorithm with parameters that are even remotely in the range where the analysis works, one would certainly end up with boostrapped value functions that are more optimistic than those of optimistic LSVI (also run with parameters dictated by the analysis).
> > I agree with the authors' response that more optimism is often detrimental to the empirical performance and only necessary optimism is desirable (after all e-greedy or greedy are very competitive baselines in many environments). Of course, in practice one would always tune parameters to reduce the optimism but the gap between parameters required by theory and those that work in practice seems much larger for BooVI than for LSVI-UCB. Also, the experiments seem to not show better performance of BooVI compared to LSVI-UCB, even with tuning.
> >
> > All in all, I see potential in this algorithm but the current results do not provide sufficient support that the algorithm can realize this potential beyond linear MDPs.
> > In light of the response and other reviews I raise my score but retain my assessment that the paper needs to be revised with additional support for the claim that this approach combines the theoretical guarantees of LSVI-UCB with the "practical advantage of randomized LSVI" in linear MDPs and beyond.

---

### Decision · Program_Chairs · 2021-09-27

**Decision:**

Accept (Poster)

**Comment:**

The paper improves the state-of-the-art analysis of randomized RL algorithms with function approximation. The authors should improve the presentation and add more empirical support.